# Interpreting Neural Networks Through the Lens of Heat Flow

## Abstract

Machine learning models are often developed in a way that prioritizes task-specific performance but defers the understanding of how they actually work. This is especially true nowadays for deep neural networks. In this paper, we step back and consider the basic problem of understanding a learned model represented as a smooth scalar-valued function. We introduce HeatFlow, a framework based upon the heat diffusion process for interpreting the multi-scale behavior of the model around a test point. At its core, our approach looks into the heat flow initialized at the function of interest, which generates a family of functions with increasing smoothness. By applying differential operators to these smoothed functions, summary statistics (i.e., explanations) characterizing the original model on different scales can be drawn. We place an emphasis on studying the heat flow on data manifold, where the model is trained and expected to be well behaved. Numeric approximation procedures for implementing the proposed method in practice are discussed and demonstrated on image recognition tasks.

## 1 Introduction

In recent years, thanks to the growing availability of computation power and data, together with the rapid advancement of methodology, the machine learning community is witnessing the success of creating models with increasingly higher capacity and performance. However, a downside of scaling the model complexity is that it complicates the understanding of how the learned models work and why sometimes they fail. Such requirements for interpretability arise from both scientific research and engineering practices. Carefully interpreting the working mechanism of a predictive model may help uncover its weakness in robustness, informing further improvements should to be made before deployment in high-stakes decision-making.

In this paper, we consider the interpretation of scalar-valued smooth functions, a basic hypothesis class in machine learning. Models of this type arise naturally in regression and binary classification tasks that deal with continuous input features. Multi-output models, e.g., neural networks for multi-class classification, can be treated as such functions by investigating each output separately. While 1D and 2D such functions can be understood intuitively through graphical visualization, there is no straightforward way to visualize or even imagine general higher-dimensional functions. Fortunately, mathematicians developed the *derivative* to interpret functions in a pointwise manner. The directional derivative at a point measures the instantaneous rate of change of the function along a given direction, and the gradient gives the direction of steepest ascent. This forms the basis of popular gradient-based explanation methods for neural networks (Simonyan et al., 2014; Selvaraju et al., 2017; Sundararajan et al., 2017; Smilkov et al., 2017; Ancona et al., 2018; Erion et al., 2021; Xu et al., 2020; Hesse et al., 2021; Srinivas & Fleuret, 2021; Kapishnikov et al., 2021).

To interpret the outcome of a learned high-dimensional function at a test point, the gradient there is only part of the story because it just characterizes the first-order behavior of the function in an infinitesimal range. Such extreme localness is to blame for several known pitfalls of vanilla gradient-based interpretation. For example, if the point falls into a locally constant region, the gradient will be zero (Shrikumar et al., 2017). On the other hand, the gradient may change dramatically even for nearby points, leading to noisy and non-robust explanations in practice (Dombrowski et al., 2019; Wang et al., 2020). Moreover, the gradient will also be zero at different classes of critical points, suggesting the need for higher-order derivatives.

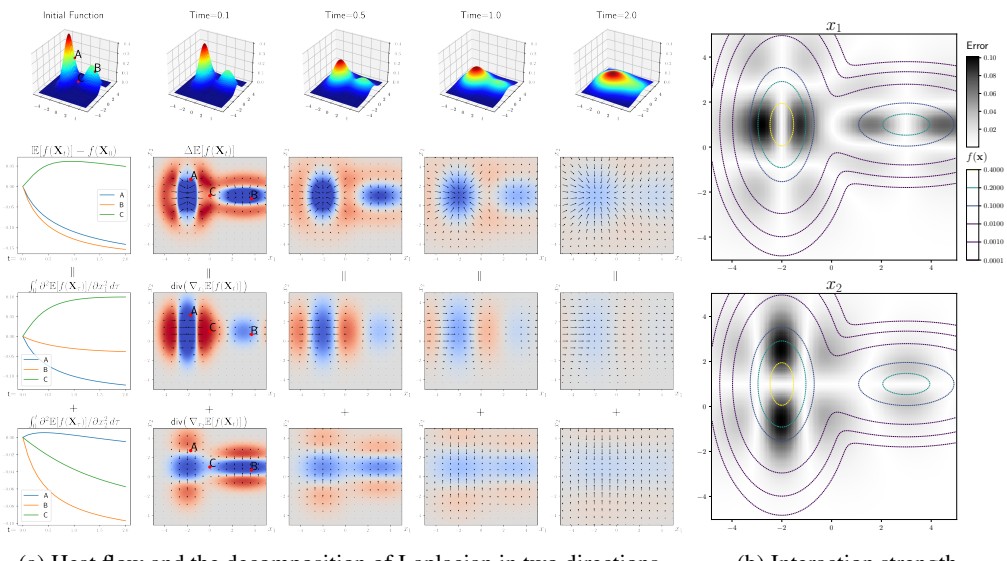

(a) Heat flow and the decomposition of Laplacian in two directions.  (b) Interaction strength.

Figure 1: A toy example in $\mathbb{R}^2$.

To this end, we introduce HeatFlow, an interpretation framework that enables summarizing the behavior of a learned model at different scales from the point of view of a test input. Our approach is motivated by a natural question to ask about a function $f$: How much does the value of $f$ at a point $\mathbf{x}$ deviate from the average value of $f$ in a neighborhood of $\mathbf{x}$? In our opinion, such deviation from local average is more comprehensible than instantaneous rate of change for non-mathematical audience. If the neighborhood is taken to be a small open ball centered at $\mathbf{x}$, then the answer is related to $\Delta f(\mathbf{x})$, where $\Delta$ is the Laplace operator, a fundamental second-order differential operator. To consider increasingly larger neighborhoods in a multi-scale manner, we propose solving a *heat equation*, a fundamental partial differential equations (PDE) studied in mathematics and physics.

We show that detailed interpretation for a function of interest may be achieved by extracting a rich set of principled summary statistics from the solution of the heat equation initialized at it. Furthermore, because a learned function is expected to be well-behaved only on the data manifold embedded in Euclidean space, it is possible to restrict the function on the manifold and solve the corresponding heat equation. In doing so, the interpretation problem is treated in a principled way based on the theory of differential geometry. Briefly, HeatFlow satisfies the following desiderata: **(i)** It provides a **multi-scale** analysis of feature importance in the formation of model predictions. **(ii)** It is **stable** and **informative** because, implicitly, the neighborhood of a test point is exhaustively explored by Brownian motion. **(iii)** It offers practitioners the **flexibility** to restrict their analysis on a manifold chosen from among Euclidean space, (learned) data manifold, and other interested submanifolds.

**A Toy Example.** A toy example of understanding the sum of two Gaussian functions in a 2D Euclidean space is illustrated in Figure 1a. First row demonstrates the initial function and its heat flow. The heat flow generates a sequence of functions that are increasingly smoother than the initial one. Later we will see one possible representation of these functions as $\mathbb{E}[f(X_t)|X_0 = \mathbf{x}]$, in which $f$ is the initial function and $\{X_t\}_{t>0}$ is a path of Brownian motion. Intuitively, the values of these smoothed functions at $\mathbf{x}$ are local average values of the initial function in increasingly larger neighborhoods centered at $\mathbf{x}$. The first subplot in the second row shows the deviation between the initial function and smoothed functions at three points, while the remaining subplots show Laplacian of smoothed functions along with their gradient fields. The deviation and the Laplacian are further decomposed into two directions, $x_1$ and $x_2$, presented in the third and forth rows, respectively. Intuitively, our core idea is to distribute the deviation caused by the heat flow to each input feature by decomposing the heat flow as the sum of "sub-flows" in corresponding directions. We show that the proposed method further enables the detection of interaction strength between one variable and other variables, as demonstrated in Figure 1b.

## 2 PRELIMINARIES

In this section, we introduce some basic concepts of explanation methods (Covert et al., 2021; Arrieta et al.) in the interpretable AI community, along with definitions and key concepts about the Laplacian and the heat equation (Hsu, 2002; Grigoryan, 2009). The connections between the two fields where this paper is inspired are also presented. Throughout our discussion, we consider a closed and connected manifold $\mathcal{M}$ endowed with a Riemannian metric $g$. A function $f : \mathcal{M} \to \mathbb{R}$ is assumed to be bounded and twice continuously differentiable.

### 2.1 FEATURE IMPORTANCE & ATTRIBUTION

Let $f : \mathbb{R}^D \to \mathbb{R}$ be a trained neural network that predicts a label $y \in \mathbb{R}$ for a given input $\mathbf{x} = (x_1, ..., x_D) \in \mathbb{R}^D$. Post hoc interpretation methods aim at generating human-friendly explanations to explain why $y = f(\mathbf{x})$. One of the most popular types of explanations is *feature importance*, which assigns certain score to each feature $x_i$ of a particular data point $\mathbf{x}$. For differentiable $f$, the *gradient* $\nabla f : \mathbb{R}^D \to \mathbb{R}^D$ is the simplest method that quantifies local importance of each input feature to the model's outcome. *Attribution* methods go a step further to assign each feature a value $\psi_i(f, \mathbf{x})$ with the same physical meaning and unit of measurement as the model's output. It is then enforced that the values for all input features sum up to the difference between the outcome $y$ and a baseline $b$, i.e., $f(\mathbf{x}) - b = \sum_{i=1}^{D} \psi_i(f, \mathbf{x})$. Previous work usually chose the baseline as the model prediction on a *baseline input* $\mathbf{x}'$, i.e., $b := f(\mathbf{x}')$. However, the choice of baseline input is known to be non-trivial (Sturmfels et al., 2020). In this work, we directly choose the baseline $b$ instead and design a corresponding attribution method. An interesting setting of $b$ is the locally averaged outcome of $f$ in the neighborhood around $\mathbf{x}$, that is, $b := \mathbb{E}_{X \sim \mathcal{N}(\mathbf{x})}[f(X)]$. To avoid selecting a single neighborhood distribution $\mathcal{N}(\mathbf{x})$, we further suggest, informally, considering a sequence of increasingly larger neighborhoods $\{\mathcal{N}_t(\mathbf{x})\}_{t \geq 0}$, where we expect $t$ to govern the scale of neighborhood. To obtain this sequence, we turn to Laplacian and heat equations introduced below.

The *manifold hypothesis* assumes that data in real world concentrates on a low-dimensional submanifold $\mathcal{M}$ embedded in the ambient space $\mathbb{R}^D$ of much higher dimension. The function to be explained is expected to behave consistently only on $\mathcal{M}$ despite being learned in $\mathbb{R}^D$. For a learned function $f : \mathbb{R}^D \to \mathbb{R}$, we may be interested in its mechanism in either the open world or a closed world. Arbitrary input in $\mathbb{R}^D$ will be allowed in the former setting, while the latter setting only accepts input on the data manifold. Thus, users should choose the appropriate target and manifold accordingly depending on the purpose of model interpretation. If the goal is to understand model behavior in the open world, $f$ and $\mathbb{R}^D$ should be chosen; while $f_{|\mathcal{M}}$, the restriction of $f$ on $\mathcal{M}$, and $\mathcal{M}$ are suggested if the analysis is restricted to the data manifold.

### 2.2 THE LAPLACIAN ON A RIEMANNIAN MANIFOLD

Since we are interested in manifolds apart from Euclidean space, we introduce the *Laplace-Beltrami* operator, $\Delta_{\mathcal{M}}$, which generalizes the ordinary Laplacian to Riemannian manifold. There are multiple equivalent ways of introducing this operator (Hsu, 2002, Section 3.1).

- Divergence of the gradient field: $\Delta_{\mathcal{M}} f = \operatorname{div} \operatorname{grad} f$. The gradient $\operatorname{grad} f$ is the unique vector field on $\mathcal{M}$ that satisfies $\langle \operatorname{grad} f, X \rangle_g = \operatorname{d}f(X)$ for any vector field $X$, where the differential $\operatorname{d}f$ gives the directional derivative of $f$ along $X$. The divergence $\operatorname{div} X$ of a vector field $X$ measures how much it locally behaves like a sink or source. Laplacian will therefore be positive at minima and negative at maxima. More generally, it acts as a measure of deviation from local average. The relevance of Laplacian in our work is then obvious: through a closer look into Laplacian, it is possible to show, for example, why the model's output at a local maxima(minima) is larger(less) than its neighborhood.

- Trace of the Hessian: $\Delta_{\mathcal{M}} f = \operatorname{tr}(\nabla^2 f)$. In Euclidean space $\mathbb{R}^n$, Hessian $\nabla^2 f$ is a matrix of the second partial derivatives, hence $\Delta f = \sum_i \frac{\partial^2}{\partial x_i^2} f$. In Riemannian manifold with its Levi-Civita connection $\nabla$, Hessian is the second covariant derivative, such that given local coordinates $\{x^i\}$, a local expression for the Hessian tensor is $\nabla^2 f = \left( \frac{\partial^2 f}{\partial x^i \partial x^j} - \Gamma_{ij}^k \frac{\partial f}{\partial x^k} \right) dx^i \otimes dx^j$, where $\Gamma_{ij}^k$ are the Christoffel symbols of connection. Therefore the Laplacian contains information about the second order behavior of a function, which is not available in gradients.

- Hodge Laplacian: $\Delta_{\mathcal{M}} f = -\mathrm{d}^* \mathrm{d} f$. In exterior calculus, the codifferential $\mathrm{d}^*$ is adjoint to the differential $\mathrm{d}$ and by $\int f d^* \theta dV_g = \int \langle df, \theta \rangle_g dV_g$, where $V_g$ is the Riemannian volume. When acting on 1-forms, $\mathrm{d}^*$ can be given by the divergence as $\mathrm{d}^*(\alpha_X) = -\operatorname{div}(X)$ where $\alpha_X$ is the dual of $X$ by $\alpha_X(Y) = \langle X, Y \rangle_g$ and hence $\Delta_{\mathcal{M}} f = \operatorname{div} \operatorname{grad} f = -\mathrm{d}^* \mathrm{d} f$. The *Hodge* Laplacian for differential forms, given by $\square_{\mathcal{M}} = -(\mathrm{d}^* \mathrm{d} + \mathrm{d} \mathrm{d}^*)$, coincides with $\Delta_{\mathcal{M}} f$ when acting on functions, i.e. 0-forms. Generalization of Laplacian to differential forms allows us to analysis the gradient field of a function in addition to the function itself.

- Infinitesimal generator that generates Brownian motion on $\mathcal{M}$: roughly speaking, it means $\frac{1}{2} \Delta_{\mathcal{M}} f(x) = \lim_{t \downarrow 0} \frac{1}{t} (\mathbb{E}[f(X_t)|X_0 = \mathbf{x}] - f(\mathbf{x}))$, where $\{X_t\}_{t>0}$ is a Brownian path. This connection provides a clue to the definition of a distribution on the neighborhood of $\mathbf{x}$ through Brownian motion.

## 2.3 HEAT EQUATION

With the above Laplacian defined, we can now introduce the most basic diffusion process governed by the following PDE, which describes how an initial heat distribution $f : \mathcal{M} \to \mathbb{R}$ looks after being diffused for time $t > 0$:

$$\frac{\partial}{\partial t} u(t, \mathbf{x}) = \Delta_{\mathcal{M}} u(t, \mathbf{x}), \tag{1}$$

$$u(0, \mathbf{x}) = f(\mathbf{x}). \tag{2}$$

The solution, $u(t, \mathbf{x}) : (0, \infty) \times \mathcal{M} \to \mathbb{R}$, can also be given in multiple roughly equivalent ways:

- Convolution: $u(t, \mathbf{x}) = \int_{\mathcal{M}} k_t(\mathbf{x}, \mathbf{y}) f(\mathbf{y}) d\mathbf{y}$, in which $k_t(\mathbf{x}, \mathbf{y})$ is the fundamental solution of this PDE, known as the *heat kernel*. In Euclidean space, it is just a Gaussian centered at the point $\mathbf{x}$.

- Expectation: $u(t, \mathbf{x}) = \mathbb{E}[f(X_t)|X_0 = \mathbf{x}]$. As the heat kernel is also the transition density function of Brownian motion on $\mathcal{M}$, the expectation of function value over ends of Brownian paths yields the solution on a stochastically complete manifold (Hsu, 2002). Such a stochastic representation of PDE solutions is known as the *Faynman-Kac* formula (Karatzas & Shreve, 2012).

- Gradient flow (Santambrogio, 2016): It is well-known that the solution is the gradient flow for the Dirichlet energy $\int \| \operatorname{grad} f \|_g^2 dV_g$ in $L^2(\mathcal{M})$ space. Consequently, the Dirichlet energy monotonically decreases in time under the heat flow, meaning that the smoothness of the solution is always increasing.

It is convenient to define the *heat operators* $\{P_t\}_{t>0}$ to represent the solution: $u(t, \mathbf{x}) = (P_t f)(\mathbf{x})$. The heat equation can be extended to the diffusion of tensor fields and differential forms using generalized Laplacian. For instance, heat equation on 1-forms may be defined with the Hodge Laplacian: $\frac{\partial}{\partial t} \theta(t, \mathbf{x}) = \square_{\mathcal{M}} \theta(t, \mathbf{x})$. Since $\square_{\mathcal{M}}$ commutes with the differential $\mathrm{d}$, if the initial condition is set to be a closed 1-form, i.e., the differential of a function: $\theta(0, \mathbf{x}) = \mathrm{d} f(\mathbf{x})$, then the solution will have an interesting connection with the heat equation on functions (Hsu, 2002, Section 7.2):

$$\theta(t, \mathbf{x}) = \mathrm{d}(P_t f)(\mathbf{x}). \tag{3}$$

It means that diffusing the differential of a function with Hodge Laplacian is equivalent to applying the differential operator to the solution of scalar heat equation.

## 3 MULTI-SCALE INTERPRETATION BASED ON HEAT DIFFUSION

By treating the model to be explained as the initial heat distribution, we analyze the solution of the heat equation to summarize the model at different scales from the point of view of an input $\mathbf{x}$. The solution alone has already given local weighted average value of the function centered at $\mathbf{x}$: $(P_t f)(\mathbf{x}) = \mathbb{E}[f(X_t)|X_0 = \mathbf{x}]$. The time $t$ plays the role of vicinity quantification that defines the range of neighborhood through a stochastic view as the area reachable by Brownian motion starting at $\mathbf{x}$. For small value of $t$, the smoothed value is mainly affected by small neighborhood of $\mathbf{x}$, hence reflecting only local information of the explained model. While as $t$ increases, the smoothed function tends to capture more global trends of the model. In this section, we show that it is possible to extract more detailed information from the solution by applying differential operators to $\{P_t f\}_{t>0}$.

### 3.1 Summarizing Gradients

By Eq. 3 and the duality of differential and gradient, taking the gradient of $P_t f$ is equivalent to smoothing the gradient field of $f$ through running a heat equation for vector fields up to time $t$. In Euclidean settings, the gradient of $P_t f$ is $\nabla P_t f(\mathbf{x}) = \mathbb{E}[\nabla f(X_t)|X_0 = \mathbf{x}]$, which is simply the locally averaged version of the initial gradient field. On more general manifolds, it does not make sense to add vectors living in different tangent spaces $T_{X_t}\mathcal{M}$, and we need the parallel transport (Lee, 2018, Chapter 4) term acting as a mechanism to connect nearby tangent spaces. There is a Feynman-Kac formula for this operation (Hsu, 2002, Theorem 7.2.1):

$$\operatorname{grad} P_t f(\mathbf{x}) = \mathbb{E}[M_t \tau_t \operatorname{grad} f(X_t)|X_0 = \mathbf{x}], \tag{4}$$

where $\tau_t$ is the stochastic parallel transport map that transfers vectors from the tangent space of $X_t$ back to that of $X_0$ along the Brownian path, and $M_t$ is related to the Ricci curvature tensor on $\mathcal{M}$. One can interpret Eq. 4 as collecting derivative information from neighborhood by sending Brownian particles that travel for a fixed time length, parallel transporting the gradients at the positions of particles back to the explained point and averaging them.

### 3.2 Attribution

For attribution tasks, a natural baseline to choose in our framework is the local average $\mathbb{E}[f(X_t)|X_0 = \mathbf{x}]$. Formally, given a model function $f : \mathcal{M} \subset \mathbb{R}^D \to \mathbb{R}$, the input of interest $\mathbf{x}$, and the solution $P_t f$ of heat equation initialized at $f$, we attribute $(P_t f)(\mathbf{x}) - f(\mathbf{x})$ to input features $\{x_i\}_{i=1}^D$ at each $t$.

In order to allocate the difference to the coordinates of ambient space $\mathbb{R}^D$, we try to disaggregate the Laplace-Beltrami operator using the the standard orthonormal basis $\{\mathbf{e}_i\}_{i=1}^D$. Let $\mathcal{E}_i(\mathbf{x})$ be the orthogonal projection of the unit vector $\mathbf{e}_i$ onto the tangent space $T_\mathbf{x}\mathcal{M}$. Since the gradient field is tangential to $\mathcal{M}$, we have $\operatorname{grad} f = \sum_i \langle \operatorname{grad} f, \mathbf{e}_i \rangle_g \mathbf{e}_i = \sum_i \langle \operatorname{grad} f, \mathcal{E}_i \rangle_g \mathcal{E}_i$. The projection of $\operatorname{grad} f$ onto the $i$th dimension is achieved by $\operatorname{grad}_i f = \langle \operatorname{grad} f, \mathcal{E}_i \rangle_g \mathcal{E}_i = (\mathcal{E}_i f)\mathcal{E}_i$, and we define the dual form of $\operatorname{grad}_i f$ as $d_i f$ to be the "partial differential" operator. Further, we can compute the contribution of each feature $i$ by taking the divergence of the projected vector field, $\operatorname{div}((\mathcal{E}_i f)\mathcal{E}_i)$. In this representation, the flow of heat along the $i$th dimension for an input $\mathbf{x}$ up to time $T$ is defined as,

$$\operatorname{HeatFlow}_i^\mathcal{M}(\mathbf{x}, T, f) := \int_{t=0}^T \operatorname{div} \operatorname{grad}_i (P_t f)(\mathbf{x}) \, dt. \tag{5}$$

A desirable property of HeatFlow is that the attributions add up to the difference between the values of $f$ and its smoothed version as follows,

$$(P_T f)(\mathbf{x}) - f(\mathbf{x}) = \int_{t=0}^T \frac{\partial}{\partial t}(P_t f)(\mathbf{x}) dt = \int_{t=0}^T \Delta_\mathcal{M}(P_t f)(\mathbf{x}) dt = \sum_{i=1}^D \operatorname{HeatFlow}_i^\mathcal{M}(\mathbf{x}, T, f), \tag{6}$$

where the last equality is an immediate result of $\Delta_\mathcal{M} f = \operatorname{div} \operatorname{grad} f = \sum_{i=1}^D \operatorname{div}(\sum_i \langle \operatorname{grad} f, \mathcal{E}_i \rangle_g \mathcal{E}_i)$. Instead of computing $\operatorname{div}$ with respect to manifold, we can also globally express both $\operatorname{div}$ and $\operatorname{grad}$ using the standard basis $\{\mathbf{e}_i\}$, i.e., $\operatorname{grad} f = \sum_{i=1}^D (\mathcal{E}_i f)\mathbf{e}_i$ and $\operatorname{div} X = \sum_{i=1}^D \mathcal{E}_i X_i$ where $X_i$ is the $i$-th component of vector field $X$ expressed in ambient coordinates. This leads to another decomposition $\Delta_\mathcal{M} f = \sum_{i=1}^D \mathcal{E}_i^2 f$. Furthermore, through the trace of the Hessian expression, we also have $\Delta_\mathcal{M} f = \sum_{i=1}^D \nabla^2 f(\mathcal{E}_i, \mathcal{E}_i)$ (Hsu, 2002, Corollary 3.1.5). Together, we have the following three decompositions,

$$\sum_{i=1}^D \int_{t=0}^T \operatorname{div} \operatorname{grad}_i (P_t f)(\mathbf{x}) \, dt = \sum_{i=1}^D \int_{t=0}^T \mathcal{E}_i^2 (P_t f)(\mathbf{x}) \, dt = \sum_{i=1}^D \int_{t=0}^T \nabla^2 (P_t f)(\mathcal{E}_i(\mathbf{x}), \mathcal{E}_i(\mathbf{x})) \, dt. \tag{7}$$

It is worth noticing that while they are the same when summed over $D$ dimensions, the three summands for each dimension $i$ are not necessarily equal to each other. For simplicity of calculation, we adopt the last decomposition strategy based on the Hessian tensor in our experiments.

In Euclidean space, all the three decompositions just reduce to sum of the diagonal terms of the Hessian matrix. Hence, attribution for $i$-th dimension becomes integral of the second partial derivative:

$$\operatorname{HeatFlow}_i^{\mathbb{R}^D}(\mathbf{x}, T, f) := \int_{t=0}^T \frac{\partial^2}{\partial x_i^2}(P_t f)(\mathbf{x}) \, dt. \tag{8}$$

### 3.3 Properties in Euclidean Space

In this section, some nice properties satisfied by our method in Euclidean space are discussed. All proofs are given in the Appendix. We leave the study of analogous properties on general manifolds for future work.

**HeatFlow obeys attribution axioms.** The evolvement of the model interpretation problem brings several desirable properties that a new attribution method should satisfy (Lundberg & Lee, 2017; Sundararajan et al., 2017; Friedman, 2004). The following proposition includes four axioms defined in (Sundararajan & Najmi, 2020).

**Proposition 1.** $\text{HeatFlow}^{\mathbb{R}^D}$ *satisfies the Dummy, Efficiency, Symmetry, and Linearity axioms.*

**HeatFlow recovers GAM.** Generalized additive models (Hastie & Tibshirani, 2017; Lou et al., 2012) are a family of inherently interpretable models based on the sum of univariate functions. The interaction between the features is simply additive in a GAM, therefore a user can interpret the contribution of each feature independently through looking into the corresponding univariate function. Ideally, if an attribution method is applied to a GAM, its output is expected to be consistent with the GAM itself. This is the case for HeatFlow.

**Proposition 2.** *Suppose* $f : \mathbb{R}^D \to \mathbb{R}; \ \mathbf{x} \mapsto \sum_{i=1}^D f_i(x_i)$ *is a smooth additive function, in which* $\forall i \in \{1, \dots, D\}$, $f_i : \mathbb{R} \to \mathbb{R}$ *is a bounded continuous function in* $L^1(\mathbb{R})$. *Then* $\lim_{t \to \infty} \text{HeatFlow}_i^{\mathbb{R}^D}(\mathbf{x}, f, t) = -f_i(x_i)$.

**HeatFlow detects additive structure.** For a given neural network $f$, it is usually impossible to decompose $f$ into sum of univariate functions. How do we know if at least some variable, say, $x_i$, contributes additively to $f$? One idea is to look into the its gradient $\nabla f : \mathbb{R}^D \to \mathbb{R}^D$. Since Euclidean space is the focus of this section, let us slightly abuse the notations $\mathrm{d}f = \mathrm{grad}\, f = \nabla f$, $\mathrm{d}_i f = \mathrm{grad}_i f = [\nabla f]_i \mathbf{e}_i$, $\partial_i f = [\nabla f]_i$, and $\mathrm{d}^* X = -\mathrm{div}\, X$ for simplicity. If $x_i$ is truly contributing to $f$ additively, i.e., $\exists f_i : \mathbb{R} \to \mathbb{R}$ such that $f(\mathbf{x}) = f_i(x_i) + f_{-i}(\mathbf{x}_{-i})$, then the magnitude of the gradient projected onto the $i$-th direction can be fully characterized by $f_i$ via $\partial_i f = \mathrm{d}f_i$. The following lemma shows that the converse is also true.

**Lemma 1.** *Assume that there exists an univariate function* $f_i : \mathbb{R} \to \mathbb{R}$ *such that* $\forall \mathbf{x} \in \mathbb{R}^D :$ $\mathrm{d}f_i(x_i) = \partial_i f(\mathbf{x})$. *Then* $f(\mathbf{x}) = f_i(x_i) + f_{-i}(\mathbf{x}_{-i})$, *in which the function* $f_{-i} : \mathbb{R}^{D-1} \to \mathbb{R}$ *does not depend on* $x_i$.

Based on this observation, in order to detect if $x_i$ contributes univariately, one can try to find a function $g_i : \mathbb{R}^D \to \mathbb{R}$ whose gradient matches $d_i f$ everywhere. If such a $g_i$ exists, it will depend only on $x_i$ because its partial derivatives with respect to $\{x_j\}_{j \neq i}$ will always be zero. A straightforward implementation of this idea is to solve the optimization problem:

$$\min_{g : \mathbb{R}^D \to \mathbb{R}} |\mathrm{d}g - \mathrm{d}_i f|^2 := \int \|\mathrm{d}g(\mathbf{x}) - \mathrm{d}_i f(\mathbf{x})\|_2^2 \, d\mathbf{x} \tag{9}$$

It is well known that the Euler–Lagrange equation for this problem is the Poisson equation $\Delta g = \mathrm{d}^* \mathrm{d}_i f$. Interestingly, we discover that HeatFlow is solving this kind of equation.

**Proposition 3.** $g_{i,t}(\cdot) = \text{HeatFlow}_i^{\mathbb{R}^D}(\cdot, f, t)$ *solves the Poisson equation* $\Delta g_{i,t} = \mathrm{d}^* \mathrm{d}_i(P_t f - f)$.

As a result, for a function $f$ satisfying $\forall \mathbf{x} : \lim_{t \to \infty} P_t f(\mathbf{x}) = 0$, e.g., $f \in L^1(\mathbb{R}^D)$, the difference between the gradient of HeatFlow at large enough $t$ and the partial gradient of $-f$ approximately measures the degree of deviation from being univariate for a considered feature. Specifically, if the feature $i$ rarely interacts with other features, the following residual should be small everywhere:

$$r_i(\mathbf{x}) = \left\| d\big( \text{HeatFlow}_i(\mathbf{x}, f, t)\big) - d_i(P_t f - f)(\mathbf{x}) \right\|_2^2. \tag{10}$$

When this residual is zero everywhere, according to Lemma 1, the feature $i$ contributes to $P_t f - f$ additively. In Figure 1b, we visualize the residuals at a large $t$ for the toy example. Interestingly, similar residuals are also discovered in a different setting where the Shapley value (Shapley, 1953) is employed for attribution (Kumar et al., 2021).

## 4 CONNECTION WITH EXISTING WORK

Close connections can be drawn between our method and many existing attribution approaches. We discuss in this section how our method generalizes the literature of attributing prediction of a model $f(\cdot)$ on an input $\mathbf{x}$ to its features.

- **Vanilla Gradient (Grad)** (Simonyan et al., 2014). The *Vanilla Gradient* is defined as $\text{Grad}(x) = \nabla_{\mathbf{x}} f(\mathbf{x})$. It only characterizes first-order behavior of the model in an infinitesimal range.

- **Smooth Gradient (SG)** (Smilkov et al., 2017). Given a user-defined variance $\sigma$, the *Smooth Gradient* is defined as $\text{SG}(\mathbf{x}) = \mathbb{E}_{\mathbf{z} \sim \mathcal{N}(\mathbf{x}, \sigma^2 \mathbf{I})} \nabla_{\mathbf{z}} f(\mathbf{z})$. It is a special case of our method in Euclidean settings: $\text{SG}(\mathbf{x}) = \nabla_{\mathbf{x}} [(f * \mathcal{N}(0, \sigma^2 \mathbf{I}))(\mathbf{x})] = \nabla_{\mathbf{x}} u(t = \sigma^2/2, \mathbf{x})$.

- **Integrated Gradients (IG)** (Sundararajan et al., 2017). Given a user-defined baseline input $\mathbf{x}'$, IG of feature $i$ is defined as $\text{IG}_i(\mathbf{x}, \mathbf{x}') = (x_i - x_i') \cdot \int_0^1 \frac{\partial}{\partial x_i} f(\mathbf{x}' + \alpha(\mathbf{x} - \mathbf{x}')) d\alpha$, where gradients are accumulated along the straight-line path in Euclidean space. Our method can also be viewed as expectation over multiple path integrals: $\mathbb{E}[f(X_t) | X_0 = \mathbf{x}] = \mathbb{E}[\int_{X_{[0,t]}} df | X_0 = \mathbf{x}]$.

- **Expected Gradients (EG)** (Erion et al., 2019). By introducing a distribution of baselines $D$, EG is the expectation of path integrals defined as $\text{EG}_i(\mathbf{x}, D) = \mathbb{E}_{\mathbf{x}' \sim D} \text{IG}_i(\mathbf{x}, \mathbf{x}')$. When a Gaussian baseline is adopted as $D = \mathcal{N}(\mathbf{x}, \sigma^2 \mathbf{I})$, it coinsides with our method in Euclidean settings, explaining deviation from local average as $\sum_i \text{EG}_i(\mathbf{x}, D) = f(\mathbf{x}) - \mathbb{E}_{\mathbf{x}' \sim \mathcal{N}(\mathbf{x}, \sigma^2 \mathbf{I})} f(\mathbf{x}')$. When the training data itself is used, $D = D_{\text{data}}$, it coinsides with our method in manifold setting when $t \to \infty$, explaining the difference from global average of all data.

- **Blur Integrated Gradients (BlurIG)** (Xu et al., 2020). *BlurIG* extends IG by considering the path of successively blurring the input image using Gaussian filter. It solves a heat equation in the 2D plane of a single image as contrast to data space in our method.

It is worth noting that the heat equation has also found applications in diffusion-based generative modeling (Sohl-Dickstein et al., 2015; Song et al., 2021; De Bortoli et al., 2022), in which certain design of forward diffusion process is equivalent to solving the heat equation (as a special case of the Fokker–Planck equation) initialized at $p(\mathbf{x})$, the probability density function of data distribution.

## 5 EXPERIMENTS

To illustrate our multi-scale attribution methods, we demonstrate a sequence of attribution maps at discretized time steps on three image recognition tasks, including a synthetic image regression task, MNIST classification and facial age estimation[1]. We compare our method, HeatFlow, with four other methods, including **Grad**, **IG**, **SG** and **BlurIG**. As for SG, increasing levels of Gaussian noises are applied and resulting gradients are averaged over 100 samples. We denote noise level $s$ as the amount of noise to add to the input as fraction of the total spread, $\max \mathbf{x} - \min \mathbf{x}$. As for BlurIG, partial accumulation of gradients along the path is calculated, where $\alpha$ denotes variance of the $2D$ Gaussian kernel. A detailed discussion and pseudo-algorithm of the numerical implementation of HeatFlow in an end-to-end framework is presented in Appendix A.5. In the following experiments, all data manifold is learnt by VAEs. Since our method involves many differentiation operations, experiments[2] are implemented using the JAX package (Bradbury et al., 2018) with Tesla V100 GPU. Further information, such as latent dimension of VAEs and image sizes are summarized in Appendix A.7. For all attribution maps of following figures, red and blue pixels denote positive and negative values, respectively, and deeper color denotes higher absolute value and stronger contribution.

### 5.1 SYNTHETIC EXPERIMENTS FOR HIERARCHICAL FEATURE STRUCTURE

To study the ability of HeatFlow in uncovering different features with global or local effect, we design a synthetic dataset with hierarchical feature structure. A latent code $\mathbf{z} = [z_0, \cdots, z_{d-1}] \in [\text{Uniform}(0, 2\pi)]^d$ is first sampled, based on which, labels and input images are generated as shown in Figure 2a. The underlying manifold is known, which is the Cartesian products of six circles. Local features are those pixels close to the upper-left corner, while global ones are those of larger squares.

---

[1] Face images from the UTKFace https://susanqq.github.io/UTKFace/ dataset.
[2] Source code available at: https://anonymous.4open.science/r/heat-explainer-FFD0

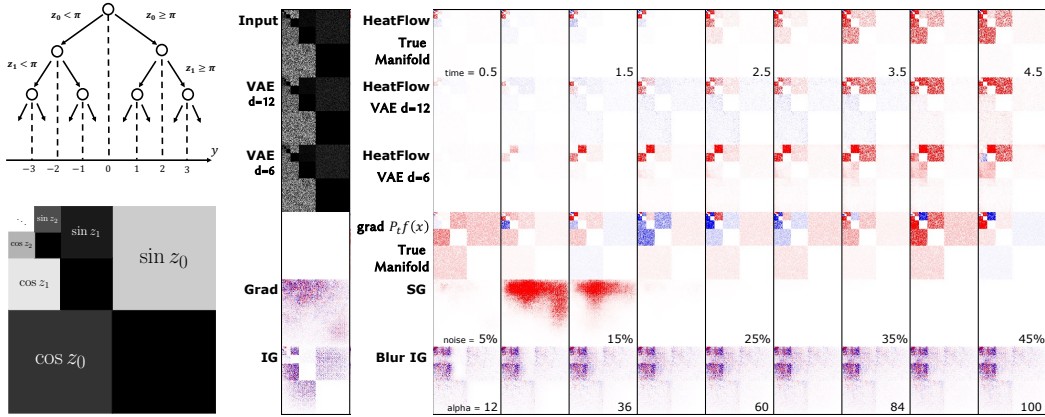

(a) Data generation.       (b) Heatmaps generated by various methods.

Figure 2: (a) **Top**: Generate labels. **Bottom**: Generate images. (b) Comparison of HeatFlow, Grad, SG, IG with black baseline and BlurIG on synthetic data. HeatFlow under true manifold, manifold learnt by VAE with latent dimension $d = 12$ and $d = 6$ are presented in the first three rows, respectively. On the fourth row, gradients on the true manifold is collected at each time step. The reconstructed inputs are shown below the original input. Vanilla Grad and IG are presented below the reconstructed inputs. For HeatFlow, SG and BlurIG, different time steps, noise levels of Gaussian blur, and partial integration up to kernel width $\alpha$ are shown, respectively.

As shown in Figure 2b, vanilla **Grad** mostly highlights local features that are close to the upper-left corner. Both **Grad** and **SG** misleadingly highlight squares lying on the diagonal which stays constant among all images. For **IG** and **BlurIG**, more global features are concentrated on, however multi-scaled information is lacking. For **HeatFlow**, localness is meticulously controlled by heat diffusion on the manifold and when value of $t$ is small, only small squares which correspond to local features are highlighted; as value of $t$ grows, more global pixels, i.e. larger squares are gradually shown. Furthermore, highlighted pixels tend to appear as an organized group in **HeatFlow**, suggesting its ability to attribute features with correlated contributions.

**Manifold Mismatch.** The heat kernel is known to be stable under perturbations of the underlying manifold (Sun et al., 2009). Intuitively, such stability comes from the Brownian motion interpretation of heat flow: small perturbations will only affect a subset of the infinitely many Brownian paths. To study the reliance of alignment between the learned manifold and the true manifold, an ablation experiment is conducted where **HeatFlow** is run separately using the true underlying manifold and manifold learned by VAE with latent dimensionality $d = 6$ and $d = 12$, shown in first three rows of Figure 2b. It is observed that, when the correctness of the manifold increases, more detailed information is captured. The multi-scale property behaves as expected regardless of degree of manifold mismatch.

## 5.2 MNIST AND UTKFACE

The same experiments are conducted on the MNIST classification task, where in Figure 3a, logits of a neural network with $98\%$ test accuracy is diffused over time. To handle all logits simultaneously, we use a heuristic extension of HeatFlow described in the Appendix. Saliency maps are particularly noisy for vanilla **Grad**. **SG** and **Blur IG** also fail to provide succinct information as Gaussian noise added in Euclidean space introduce unreliable evaluations of gradients over off-the-manifold samples. For **HeatFlow**, concise attribution with pixels only on digit area are highlighted. It is also evident that the attribution changes as heat flows. More interesting findings can be drawn if we study the logits of digits that are similar. In Figure 3b, we present heat flows and multi-scale attribution maps for three digits, $3, 6, 9$, and compare them with their visually similar digits, $\{5, 8\}, \{0, 8\}, \{7, 4\}$, respectively. It is observed that, in the last column, only pixels corresponding to the common features of the compared digits remain. At small $t$, pixels that separate digits apart are highlighted.

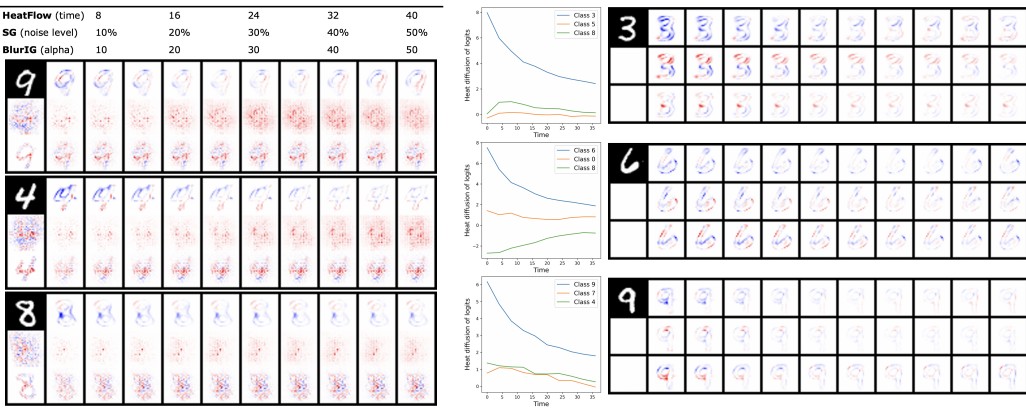

(a) Multi-scale attributions on MNIST.    (b) HeatFlow for different logits.

Figure 3: (a) Heat maps of the logit of label class on MNIST test samples. Original input is shown in the most upper-left corner. Vanilla Grad and IG are presented below the input. HeatFlow, SG and BlurIG with increasing level of time, noise and kernel width is shown on each row started from the second column, respectively. (b) Heat diffusion for MNIST samples comparing logits of different classes. **Left**: Change in function value. **Right**: HeatFlow attribution maps.

As for explaining the facial age estimation task, results are presented in Appendix A.9. Our method managed to locate structured features such as eye brows, wrinkles, whether smile with teeth shown as important information for prediction of ages, where other methods failed to discover. We also emphasize that multi-scale information is summarized in our method, such that attributed features which remains for large value of $t$ represent more global effects, such as wrinkles.

**Quantitative Evaluation.** We also include a quantitative evaluation to compare HeatFlow with other methods using a strategy adopted in (Jethani et al., 2021). This evaluation strategy utilizes the remove/retain-and-retrain ideas where an evaluator was trained to predict the label given an arbitrary subset of features, and then, performance of the evaluator was assessed as the features were gradually excluded/included according to the absolute importance output by each explanation method. Using a set of 1000 test images for MNIST and 100 for UTKFace, accuracy and mean absolute error is assessed respectively as we include or exclude the most important features ranging from $0-100\%$, with curves visualized in Figure 4. HeatFlow is very competitive in terms of AUC, indicating that it is good at identifying important discriminative features.

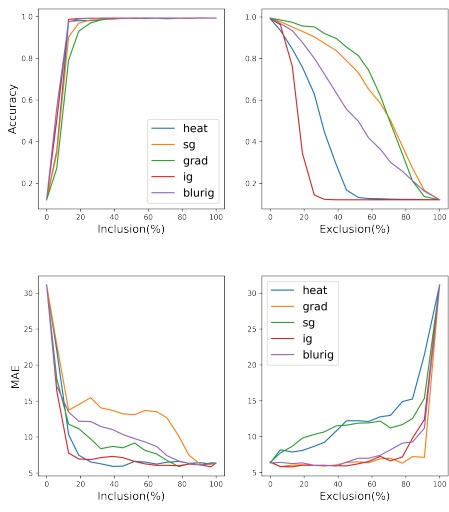

Figure 4: The change of accuracy for MNIST (upper) and MAE for UTKFace (lower) as an increasing percentage of pixels attributed to be important are included (left) or excluded (right).

## 6 CONCLUSIONS

We have introduced a novel interpretation framework, HeatFlow, which generates a sequence of feature attributions for an interested input, explaining the deviations of model's outcome from multi-scale local averages. Drawbacks of this work lie in the difficulties of manifold learning on more complex datasets, high-dimensional PDE solving, and high-order derivative computation. Our method will benefit from methodological advancements in these directions. It is also interesting to generalize HeatFlow to the union of disconnected manifolds, which may be a more appropriate assumption for classification datasets.

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

# A  Appendix

## A.1  Axiomatic Properties in Euclidean Space

*Proof of Proposition 1.*  Since the derivative of a function with respect to a dummy variable is always zero, the *Dummy* axiom is satisfied. By Eq. 6, the attributions always add up to the difference $(P_t f)(\mathbf{x}) - f(\mathbf{x})$ and hence proved the *Efficiency* axiom. The linearity is a consequence of the fact that the heat operator is linear. For a function that is symmetric in two variables $x_i$ and $x_j$, we prove below $\partial^2 P_t f/\partial x_i^2 = \partial^2 P_t f/\partial x_j^2$ as long as $x_i = x_j$.

Let $f(x_1, ..., x_i, ..., x_j, ..., x_n)$ be a function of $n$ variables, $f : \mathbb{R}^n \to \mathbb{R}$. Without loss of generality, let $x_i$ and $x_j$ be symmetric variables, i.e. $f(..., x_i, ..., x_j, ...) = f(..., x_j, ..., x_i, ...)$ for $\forall x_i, x_j \in \mathbb{R}$. Then we have

$$P_t f(..., x_i, ..., x_j, ...)$$

$$= \int_y \frac{1}{(4\pi t)^{n/2}} \exp\left( -\frac{1}{4t} \sum_{k=1}^n (x_k - y_k)^2 \right) f(..., y_i, ..., y_j, ...) d\mathbf{y}$$

$$= \int_y \frac{1}{(4\pi t)^{n/2}} \exp\left( -\frac{1}{4t} \sum_{k \notin \{i,j\}}^n (x_k - y_k)^2 + (x_i - y_i)^2 + (x_j - y_j)^2 \right) f(..., y_i, ..., y_j, ...) d\mathbf{y}$$

$$= \int_y \frac{1}{(4\pi t)^{n/2}} \exp\left( -\frac{1}{4t} \sum_{k \notin \{i,j\}}^n (x_k - y_k')^2 + (x_i - y_j')^2 + (x_j - y_i')^2 \right) f(..., y_j', ..., y_i', ...) d\mathbf{y}'$$

$$= \int_y \frac{1}{(4\pi t)^{n/2}} \exp\left( -\frac{1}{4t} \sum_{k \notin \{i,j\}}^n (x_k - y_k')^2 + (x_j - y_i')^2 + (x_i - y_j')^2 \right) f(..., y_i', ..., y_j', ...) d\mathbf{y}'$$

$$= P_t f(..., x_j, ..., x_i, ...),$$

where $\mathbf{y}'$ is the permutation of $\mathbf{y}$ by exchanging the $i$-th and $j$-th variables. The third equality follows from the change of variables formula and the fact that the determinant of a permutation matrix is $\pm 1$. Hence, we draw the conclusion that the heat operator preserves the symmetry in Euclidean space. Next, further assuming that $x_i = x_j = a$ and according to the limit for second derivative, we have,

$$\frac{\partial^2}{\partial x_i^2} f(..., a, ..., a, ...)$$

$$= \lim_{h \to 0} \frac{f(..., a+h, ..., a, ...) - 2f(..., a, ..., a, ...) + f(..., a-h, ..., a, ...)}{h^2}$$

$$= \lim_{h \to 0} \frac{f(..., a, ..., a+h, ...) - 2f(..., a, ..., a, ...) + f(..., a, ..., a-h, ...)}{h^2}$$

$$= \frac{\partial^2}{\partial x_j^2} f(..., a, ..., a, ...).$$

In other words, we have shown that the second derivatives of the symmetric function with respect to symmetric variables are equal when the values are equal. As a result, we have proved that $\partial^2 P_t f/\partial x_i^2 = \partial^2 P_t f/\partial x_j^2$ as long as $x_i = x_j$.  □

## A.2  Deviation from Additive Contribution and Link with Shapley Value

An interesting link exists between our method and the Hodge decomposition of a coorperative game (Stern & Tettenhorst, 2019). Given a set of $D$ players and a value function $v : 2^{[D]} \to \mathbb{R}$, each coalition of players $S \subset 2^{[D]} = V$ can be considered as a vertex of a $D$-dimensional hypercube $G = (V, E)$, where each edge corresponds to addition of a single player $i \notin S$ to $S$. The gradient and divergence of exterior calculus operating on this graph, denoted by $(d, d^*)$, are defined as $dv(S, S \cup \{i\}) = v(S \cup \{i\}) - v(S)$ and $(d^* f)(a) = \sum_{(b,a) \in E} f(b, a)$, respectively. $dv \in \ell^2(E)$ gives the marginal value contributed by a player joining a coalition. To specify marginal contribution of an individual player $i \in [D]$, a partial gradient $d_i : \ell^2(V) \to \ell^2(E)$ is defined as

$$d_i v(S, S \cup \{j\}) = \begin{cases} v(S \cup \{j\}) - v(S), & i = j \\ 0, & \text{otherwise} \end{cases}.$$

Notice the corresponding definition of partial differentiation operator in the continuous space adopted in our method, $d_i f = \text{grad}_i f$. A key result in (Stern & Tettenhorst, 2019) is connecting inessentiality of games to the defined partial gradient operators in the following proposition, where in an inessential game, for all $S \subseteq V$, $v(S) = \sum_{i \in S} v(\{i\})$, meaning each player contributes a precise value $v(\{i\})$ to any coalition it participates in.

**Proposition 4.** *(Stern & Tettenhorst, 2019, Prop 3.3) A game is inessential if and only if $d_i v \in \text{im} \, d$ for all $i \in [D]$.*

This means a game is inessential if one can find games $v_i$ such that $d_i v = dv_i$ for each player $i$. In our setting, there is no notion of game but we consider in breaking down the given differentiable model $f(\cdot)$ to additive univariate functions. This ability in separating each feature to contribute independently is the "inessentiality" of our problem.

In particular, we consider the simplest case where the function to be explained is itself an additive one, i.e. $f : \mathbb{R}^D \to \mathbb{R}$; $\mathbf{x} \mapsto \sum_{i=1}^{D} f_i(x_i)$. We prove Proposition 2 that our method $\text{HeatFlow}_i^{\mathbb{R}^D}(\cdot, T, f)$ tends to recover each independent component as $T$ tends to infinity.

*Proof.* The solution of heat equation initialized at an additive function is the sum of 1D solutions, as follow,

$$P_t f(\mathbf{x}) = \int \frac{1}{(4\pi t)^{D/2}} \exp\left(-\frac{1}{4t}\|\mathbf{x} - \mathbf{y}\|^2\right)\left(\sum_{i=1}^{D} f_i(y_i)\right) d\mathbf{y}$$

$$= \sum_{i=1}^{D} \int \frac{1}{(4\pi t)^{1/2}} \exp\left(-\frac{1}{4t}(x_i - y_i)^2\right) f_i(y_i) \, dy_i = \sum_{i=1}^{D} (P_t f_i)(x_i)$$

Following this observation, it is easy to derive that,

$$\text{HeatFlow}_i^{\mathbb{R}^D}(\mathbf{x}, T, f)$$
$$:= \int_{t=0}^{T} \frac{\partial^2}{\partial x_i^2}(P_t f)(\mathbf{x}) \, dt = \int_{t=0}^{T} \frac{\partial^2}{\partial x_i^2}(P_t f_i)(x_i) \, dt = \int_{t=0}^{T} \frac{\partial}{\partial t}(P_t f_i)(x_i) \, dt = P_T f_i(x_i) - f_i(x_i)$$

For $f_i \in L^1(\mathbb{R})$, $\lim_{t\to\infty} P_t f_i(x_i) = 0$. Hence, Proposition 2 is proved. $\qquad\square$

Next, we consider more general case, where some variables might interact with each other.

*Proof of Lemma 1.* Assume an arbitrary path over $\mathbb{R}^n$, $\mathbf{r}(t) = (x_1(t), \ldots, x_n(t))^T$, such that the end points are $\mathbf{r}(0) = \mathbf{x}_a$ and $\mathbf{r}(1) = \mathbf{x}_b$, particularly, the $i$-th feature has value $(\mathbf{x}_a)_i = a$ and $(\mathbf{x}_b)_i = b$. By the fundamental theorem of calculus for line integrals, and if $\frac{\partial f}{\partial x_i} = \frac{df_i}{dx_i}$ for any $\mathbf{x}$, we have,

$$f(\mathbf{x}_b) - f(\mathbf{x}_a) = \int_0^1 \frac{\partial f(\mathbf{r}(t))}{\partial \mathbf{x}} \cdot \frac{d\mathbf{r}(t)}{dt} dt$$

$$= \int_0^1 \sum_{i=1}^{n} \frac{\partial f(\mathbf{r}(t))}{\partial x_i} \frac{dx_i(t)}{dt} dt$$

$$= \sum_{j \neq i} \int_0^1 \frac{\partial f(\mathbf{r}(t))}{\partial x_j} \frac{dx_j(t)}{dt} dt + \int_0^1 \frac{\partial f(\mathbf{r}(t))}{\partial x_i} \frac{dx_i(t)}{dt} dt$$

$$= \sum_{j \neq i} \int_0^1 \frac{\partial f(\mathbf{r}(t))}{\partial x_j} \frac{dx_j(t)}{dt} dt + \int_0^1 \frac{df_i(x_i(t))}{dx_i} \frac{dx_i(t)}{dt} dt$$

$$= \sum_{j \neq i} \int_0^1 \frac{\partial f(\mathbf{r}(t))}{\partial x_j} \frac{dx_j(t)}{dt} dt + f_i(b) - f_i(a).$$

With $\mathbf{x}_a$ set to $\mathbf{0}$ and assuming $f(\mathbf{0}) = 0$, we have $f(\mathbf{x}) = \sum_{j \neq i} \int_0^1 \frac{\partial f(\mathbf{r}(t))}{\partial x_j} \frac{dx_j(t)}{dt} dt + f_i(x_i)$. Since path $\mathbf{r}(t)$ is arbitrary meaning the equation holds for any path between $\mathbf{0}$ and $\mathbf{x}$, if the first term depends on $x_i$, then by differentiating both sides with respect to $x_i$, $\frac{\partial f}{\partial x_i} = \frac{df_i}{dx_i}$ would be contradicted and hence Lemma 1 proved. $\qquad\square$

Next, we prove Proposition 3, such that our method $\text{HeatFlow}_i^{\mathbb{R}^D}(\cdot, t, f)$ is solving the Poisson equation $\Delta g_{i,t} = d^* d g_{i,t} = d^* d_i (P_t f - f)$.

*Proof.* First we emphasize that operator $d^* d_i$ and the heat operator $P_t$ commute with each other since $d^* d_i$ commute with the Laplacian $\Delta$ as follows:

$$
\begin{aligned}
d^* d_i \Delta f &= \text{div}\left( \frac{\partial}{\partial x_i} \sum_{k=1}^{D} \frac{\partial^2}{\partial x_k^2} f \right) \\
&= \frac{\partial^2}{\partial x_i^2} \sum_{k=1}^{D} \frac{\partial^2}{\partial x_k^2} f \\
&= \sum_{k=1}^{D} \frac{\partial^2}{\partial x_k^2} \frac{\partial^2}{\partial x_i^2} f = \Delta d^* d_i f.
\end{aligned}
$$

With this fact, we have

$$
\begin{aligned}
d^* d \Big( \text{HeatFlow}_i(\mathbf{x}, t, f) \Big) &= d^* d \int_{\tau=0}^{t} d^* d_i (P_\tau f)(\mathbf{x}) d\tau \\
&= d^* d \int_{\tau=0}^{t} P_\tau (d^* d_i f)(\mathbf{x}) d\tau \\
&= \int_{\tau=0}^{t} \Delta P_\tau (d^* d_i f)(\mathbf{x}) d\tau \\
&= \int_{\tau=0}^{t} \frac{\partial}{\partial t} P_\tau (d^* d_i f)(\mathbf{x}) d\tau \\
&= P_t (d^* d_i f)(\mathbf{x}) - d^* d_i f(\mathbf{x}) = d^* d_i (P_t f - f)(\mathbf{x}).
\end{aligned}
$$

$\square$

### A.3  EXAMPLE ON MULTIVARIATE GAUSSIAN

Here, we present a simple example of our method, HeatFlow, on a $d$-dimensional multivariate Gaussian with mean $\boldsymbol{\mu}$ and diagonal covariance $\boldsymbol{\Sigma} = \text{diag}(\sigma_1, \dots, \sigma_d)$. Firstly, the solution of heat equation is calculated as follow,

$$
\begin{aligned}
P_t f(\mathbf{x}) &= \int k_t(\mathbf{x}, \mathbf{y}) \mathcal{N}(\mathbf{y}; \boldsymbol{\mu}, \boldsymbol{\Sigma}) \, dy \\
&= \int \frac{1}{(4\pi t)^{d/2}} \exp\big( -\frac{1}{4t} \|\mathbf{x} - \mathbf{y}\|^2 \big) \frac{1}{(2\pi)^{d/2} |\boldsymbol{\Sigma}|^{1/2}} \exp\big( -\frac{1}{2} (\mathbf{y} - \boldsymbol{\mu})^T \boldsymbol{\Sigma}^{-1} (\mathbf{y} - \boldsymbol{\mu}) \big) \, dy \\
&= \mathcal{N}(\mathbf{x}; \boldsymbol{\mu}, \boldsymbol{\Sigma} + 2tI)
\end{aligned}
$$

Consequently, we have the contribution of the $i$-th variable as,

$$
\text{HeatFlow}_i^{\mathbb{R}^D}(\mathbf{x}, T, f) := \int_{t=0}^{T} \frac{\partial^2}{\partial x_i^2} (P_t f)(\mathbf{x}) \, dt = \int_{t=0}^{T} \frac{1}{(2\pi)^{d/2} |\boldsymbol{\Sigma} + 2tI|^{1/2}} \left( -\frac{1}{\sigma_i + 2t} \right) dt
$$

### A.4  ADDITIONAL RELATED WORK

A relevant literature from the perspective of generative models is identifying interpretable latent dimensions (Yang et al., 2021). In (Wang & Ponce, 2020), unsupervised discovery of interpretable axes is facilitated by exploring the latent space along the eigenvectors of the metric tensor defined by the decoder. Eigenvectors at different ranks encode qualitatively different type of changes. The main difference between our work and these prior literature is that our primary goal is to interpret a learned regression/classification model with the help of a generative model to explore the data manifold, while their goal is to explain the generative model itself.

The discussion in the main text assumes that we have access to a parameterized representation of the manifold $\mathcal{M}$. In practice, the manifold may be implicitly defined via a projection function that projects an arbitrary point onto the manifold through identifying its closest on-manifold counterpart in terms of Euclidean distance (Ruuth & Merriman, 2008; März & Macdonald, 2012). Given such a closest point function $\mathrm{cp}(\mathbf{x}) : \mathbb{R}^D \to \mathcal{M}$, it is possible to extend a function $f(\mathbf{x}) : \mathcal{M} \to \mathbb{R}$ to the surrounding space in $\mathbb{R}^D$ by defining a new function $f \circ \mathrm{cp}(\mathbf{x}) : \mathbb{R}^D \to \mathbb{R}$. It has been shown that the intrinsic operations of grad and $\Delta_{\mathcal{M}}$ are simplified as $\mathrm{grad}\, f(\mathbf{x}) = \nabla[f \circ \mathrm{cp}](\mathbf{x})$ and $\Delta_{\mathcal{M}} f(\mathbf{x}) = \Delta[f \circ \mathrm{cp}](\mathbf{x})$. The latter equation provides the fourth possibility to disaggregate the Laplace-Beltrami operator.

## A.5 NUMERICAL IMPLEMENTATION

In this section, we briefly discuss the numerical implementation of our method in an end-to-end framework from manifold learning to multi-scale explanation. Since there are numerous approaches for manifold learning and high-dimensional PDE solving, we claim our implementation is only one of many possible realizations and can be generalized to different manifold settings for various purposes.

**Manifold Learning.** Deep generative models strive to infer probability distribution from observational data $\mathbf{x} \in \mathcal{X} = \mathbb{R}^D$, as well as learning the underlying data manifold (Brehmer & Cranmer, 2020; Arvanitidis et al., 2018). Approaches such as VAEs (Kingma & Welling, 2014; Rezende et al., 2014) and GANs (Goodfellow et al., 2014) makes the assumption of a $d$-dimensional data manifold, $\mathcal{M}$, embedded in the ambient space with $d < D$, through highly flexible generator, $g : \mathcal{Z} \to \mathcal{M}$, where $\mathbf{z} \in \mathcal{Z} = \mathbb{R}^d$ is the latent variable. The local Jacobian of the generator function, $\mathbf{J}_{\mathbf{z}} = \frac{\partial g}{\partial \mathbf{z}}|_{\mathbf{z}=\mathbf{z}}$, provides local basis in the input space and $\mathbf{G}_{\mathbf{z}} = \mathbf{J}_{\mathbf{z}}^T \mathbf{J}_{\mathbf{z}}$ is the Riemannian metric.

**Heat Equation Solving.** To solve the heat equation on manifold, especially a high-dimensional one, we utilize the Feynman-Kac formula and the framework proposed by (Beck et al., 2021; Berner et al., 2020) where one can simulate training data in order to learn solution $P_t f$ by means of deep learning. A supervised learning problem is constructed via the predictor variables $(\mathbf{x}, t)$ and the dependent target variables $y = f(X_t)|_{X_0=\mathbf{x}}$. The unique minimizer of the quadratic loss $\min_{\phi} \mathbb{E}[(\phi(\mathbf{x}, t) - y)^2]$ is the solution of the heat equation. In practice, the following empirical error is minimized over the function space of suitable neural networks $\mathcal{H}$:

$$\hat{\phi} = \arg\min_{\phi \in \mathcal{H}} \frac{1}{s} \sum_{i=1}^{s} (\phi(\mathbf{x}_i, t_i) - f(X_t)|_{X_0=\mathbf{x}_i})^2, \tag{11}$$

where $\{(\mathbf{x}_i, t_i)\}_{i=1}^s$ are realization of i.i.d. samples uniformly drawn from $\mathcal{M} \times [0, T]$. To simulate Brownian motion on a Riemannian manifold, we perform random walks over the learned manifold depending on the Laplacian adopted. In Euclidean setting, random walks are realized by adding Gaussian noise. While in curved space, one usually resort to geodesic random walks performed directly on latent variables. Detailed algorithms are presented in the Algorithm 2.

For classification models, to align with training of the original model, we minimize the cross entropy loss, instead of MSE in Eq. 11.

**Gradient and Decomposition of Laplacian** To compute the gradient and the attribution after heat equation solving, we need to explicitly realize the projection $\mathcal{E}_i$. When a local chart $g$ is available, such as a decoder function, we define $\mathbf{J}(\mathbf{J}^T \mathbf{J})^{-1} \mathbf{J}^T$ as the projection matrix. The resulting explicit forms of $\mathrm{grad} f$ and $\mathcal{E}_i$ are

$$\mathrm{grad}\, f = (\mathbf{J}^T \mathbf{J})^{-1} \mathbf{J}^T \partial f / \partial \mathbf{x} = \mathbf{G}^{-1} \partial f / \partial \mathbf{z}, \quad \mathcal{E}_i = (\mathbf{J}^T \mathbf{J})^{-1} \mathbf{J}^T \mathbf{e}_i = \mathbf{G}^{-1} \partial g_i / \partial \mathbf{z}.$$

By the definition of Christoffel symbol which describes the metric connection of manifold, it is easy to derive $\Gamma_{jk}^i = \sum_m \frac{1}{2} \mathbf{G}_{im}^{-1}(\partial_j \mathbf{G}_{km} + \partial_k \mathbf{G}_{jm} - \partial_m \mathbf{G}_{jk}) = \sum_{m,b} \mathbf{G}_{im}^{-1} \mathbf{J}_m^b \mathbf{H}_{jk}^b$, where $\mathbf{H}_{jk}^b = \frac{\partial^2 g_b}{\partial z_j \partial z_k}$ is the Hessian matrix of the generator and $\mathbf{J}_m^b = \frac{\partial g_b}{\partial z_m}$ the Jacobian matrix. Following the definition of Hessian tensor, we have

$$\nabla^2 f(\mathcal{E}_i, \mathcal{E}_i) = \mathcal{E}_i^T \left( \frac{\partial^2 f}{\partial z_j \partial z_k} - \Gamma_{jk}^i \frac{\partial f}{\partial z_i} \right)_{jk} \mathcal{E}_i = \frac{\partial g_i}{\partial \mathbf{z}}^T \mathbf{G}^{-1} \left( \mathbf{H}_f - \mathbf{J}_f \mathbf{G}^{-1} (\sum_b \mathbf{J}^b \mathbf{H}^b) \right) \mathbf{G}^{-1} \frac{\partial g_i}{\partial \mathbf{z}}, \tag{12}$$

where $\mathbf{H}_f$ and $\mathbf{J}_f$ are the Hessian and Jacobian matrix of the explained model $f$ with respect to latent variables, respectively. Substitute this implementation into the third decomposition strategies in Eq. 7 for each $P_t f$ and then we obtain multi-scale attribution.

The following pseudo-algorithm shows a reference implementation of HeatFlow. Notice that given training input data and function to be trained, the steps of manifold learning and heat equation solving only need to be computed once. The resulting manifold and heat kernel can be reused for explaining any further test input.

---

**Algorithm 1:** A reference implementation of HeatFlow

**Require :** $\mathcal{D}$: training dataset, $f : \mathbb{R}^D \to \mathbb{R}$: a trained model, $\mathbf{x}^*$: a test input to be explained, $T$: total horizon

**Output :** $\mathbf{\Psi} = \{\psi_i^{(t)}\}_{i,t=1}^{D,T}$ feature importance for each feature $i$ and level of localness $t$

                                                  `// Manifold Learning`

1   learn the underlying manifold using VAE from $\mathcal{D}$

    **—get—:** encoder $g$, decoder $h$, Jacobian of the encoder $\mathbf{J}_g$, and metric $\mathbf{G}_g = \mathbf{J}_g^T \mathbf{J}_g$

                                           `// Heat Equation Solving`

2   initialize model $\phi$ with similar structure as $f$ and take $t$ as extra input

3   **while** *learning not done* **do**

4      **for** *each batch* **do**

5          sample $\{\mathbf{x}^{(i)}, t^{(i)}\}_{i=1}^{s}$ uniformly from $\mathcal{D} \times [0, T]$

6          **for** *each $i = 1, ...s$* **do**

7              $\mathbf{Z}^{(i)} \leftarrow \text{RW}(\mathbf{z}_0 = g(\mathbf{x}^{(i)}), s, t^{(i)}, \mathbf{G}_g)$

8          **end**

9          update $\phi$ as to minimize loss $\mathcal{L} = \sum_{i=1}^{s} \left( \phi(\mathbf{x}^{(i)}, t^{(i)}) - f(h(\mathbf{Z}^{(i)}[-1, :])) \right)^2$

10     **end**

11   **end**

    **—get—:** solution $\phi^*(\mathbf{x}, t) \approx P_t f(\mathbf{x})$ on the learned manifold

                                 `// Decomposition of Laplacian`

12   initialize $\mathbf{\Psi}$ as a zero array

13   $\mathbf{z}^* = g(\mathbf{x}^*)$

14   **for** *each $i = 1, ..., D$* **do**

15      **for** *each $t = 1, ..., T$* **do**

16          calculate $\delta \text{HeatFlow}_i = \nabla^2 \phi^*(\cdot, t)\left(\mathcal{E}_i(\mathbf{z}^*), \mathcal{E}_i(\mathbf{z}^*)\right)$ according to Eq. 12

17          accumulate $\mathbf{\Psi}[i, t] = \Psi[i, t-1] + \delta \text{HeatFlow}_i \cdot \delta t$

18      **end**

19   **end**

---

In order to simulate Brownian motion on a Riemannian manifold as needed in solving a high-dimensional heat equation, we resort to Algorithm 2, which is also adopted by (Arvanitidis et al., 2018).

---

**Algorithm 2:** Random walk on a Riemannian manifold: $\text{RW}(\mathbf{z}_0, s, T, \mathbf{G})$

**input :** Latent starting point $\mathbf{z}_0 \in \mathbb{R}^d$, step size $s$, number of steps $T$, metric tensor $\mathbf{G}$

**output :** Random walk path $\mathbf{Z} \in \mathbb{R}^{T \times d}$

1   $\mathbf{z} = \mathbf{z}_0$

2   **for** $t = 1$ *to* $T$ **do**

3      $\mathbf{L}, \mathbf{U} = \text{eig}(\mathbf{G}_\mathbf{z})$,    ($\mathbf{L}$ : eigenvalues, $\mathbf{U}$ : eigenvectors)

4      $\mathbf{v} = \mathbf{U}\mathbf{L}^{-\frac{1}{2}}\boldsymbol{\epsilon}$,   $\boldsymbol{\epsilon} \sim \mathcal{N}(\mathbf{0}, \mathbb{I}_d)$,

5      $\mathbf{z} = \mathbf{z} + s \cdot \mathbf{v}$,

6      $\mathbf{Z}(t, :) = \mathbf{z}$

7   **end**

---

A.6 DEFINITION OF MATHEMATICAL SYMBOLS

Table 1: Definition of involved math symbols, in local coordinates on a Riemannian manifold and in Cartesian coordinates in the special case of Euclidean space, along with their intuitive meanings in natural language (intuitive meaning referenced from Wikipedia). Particularly for Cartesian coordinates in Euclidean space, a three-dimensional example is also shown. Notice that the Einstein summation convention is used, implying summation over $i$ and $j$. $\mathbf{e}_i = \partial \mathbf{x} / \partial x^i$ and $\mathbf{e}^i = \mathrm{d} x^i$ refer to the unnormalized local covariant and contravariant bases, $g^{ij}$ is the inverse metric tensor, $\alpha_X$ is dual of vector field $X$, and $\langle\!\langle , \rangle\!\rangle$ is square-integrable inner product.

| Symbol | Local Coordinates | Cartesian Coordinates | Meaning |
|---|---|---|---|
| **metric tensor**: $g = \{g_{ij}\}$ | $\langle \mathbf{e}_i, \mathbf{e}_j \rangle$ | $\delta_{ij}$  identity matrix | Allow definition of distances and angles on manifolds, just as the inner product on a Euclidean space allows defining distances and angles there. |
| **gradient**: $\mathrm{grad}\, f = \nabla f$ | $\frac{\partial f}{\partial x_i} g^{ij} \mathbf{e}_j$ | $\frac{\partial f}{\partial x_i} \mathbf{e}_i$  $\begin{bmatrix} \frac{\partial f}{\partial x} & \frac{\partial f}{\partial y} & \frac{\partial f}{\partial z} \end{bmatrix}^T$ | Direction and rate of fastest increase; A tangent vector, which represents an infinitesimal change in (vector) input. |
| (dual of grad) **differential**: $\mathrm{d}f$  $\langle \nabla f, X \rangle = \mathrm{d}f(X)$ | $\frac{\partial f}{\partial x_i} \mathbf{e}^i$ | $\frac{\partial f}{\partial x_i} \mathbf{e}^i$  $\begin{bmatrix} \frac{\partial f}{\partial x} & \frac{\partial f}{\partial y} & \frac{\partial f}{\partial z} \end{bmatrix}$ | How much the (scalar) output changes for a given infinitesimal change in (vector) input. |
| **divergence**: $\mathrm{div}\, \mathbf{F} = \nabla \cdot \mathbf{F}$ | $\frac{1}{\sqrt{\det g}} \frac{\partial \sqrt{\det g} F_i}{\partial x_i}$ | $\frac{\partial F_i}{\partial x_i}$  $\frac{\partial F_x}{\partial x} + \frac{\partial F_y}{\partial y} + \frac{\partial F_z}{\partial z}$ | The volume density of the outward flux of a vector field from an infinitesimal volume around a given point. |
| (adjoint of d) **codifferential**: $\mathrm{d}^*(\alpha_X) = -\mathrm{div}(X)$  $\langle\!\langle f, \mathrm{d}^*\theta \rangle\!\rangle = \langle\!\langle \mathrm{d}f, \theta \rangle\!\rangle$ | | $-\frac{\partial F_i}{\partial x_i}$  $-\frac{\partial F_x}{\partial x} - \frac{\partial F_y}{\partial y} - \frac{\partial F_z}{\partial z}$ | The amount of "stuff" flowing through a surface locally per unit time, with velocity moving by the vector field. |
| **Laplacian**: $\Delta f = \nabla \cdot \nabla f$ | $\frac{1}{\sqrt{g}} \frac{\partial}{\partial x_i} \left( \sqrt{g} g^{ij} \frac{\partial f}{\partial x_j} \right)$ | $\frac{\partial^2 f}{\partial x_i^2}$  $\frac{\partial^2 f}{\partial x^2} + \frac{\partial^2 f}{\partial y^2} + \frac{\partial^2 f}{\partial z^2}$ | Local average deviation, how much the average value of a function over small balls centered at a point deviates from its output. |

## A.7 IMPLEMENTATION DETAILS

We summarize hyperparameter settings in our experiments in Table 2.

Table 2: Implementation details for the three experiments, synthetic, MNIST and UTKFace.

| Hyperparameter/Dataset | Synthetic | MNIST | UTKFace |
|---|---|---|---|
| Number of generated training samples | | 60000 | |
| Number of random walk steps | | 5000 | |
| Image size | $128 \times 128$ | $28 \times 28$ | $200 \times 200 \times 3$ |
| VAE latent dimension $d$ | 6/12 | 10 | 16 |
| Step size $s$ | 0.1 | 0.02 | 0.05 |
| Explained model performance | MSE$< 10^{-4}$ | ACC$> 98\%$ | MAE$< 3.0$ |

## A.8 MORE RESULTS FOR SYNTHETIC EXPERIMENTS

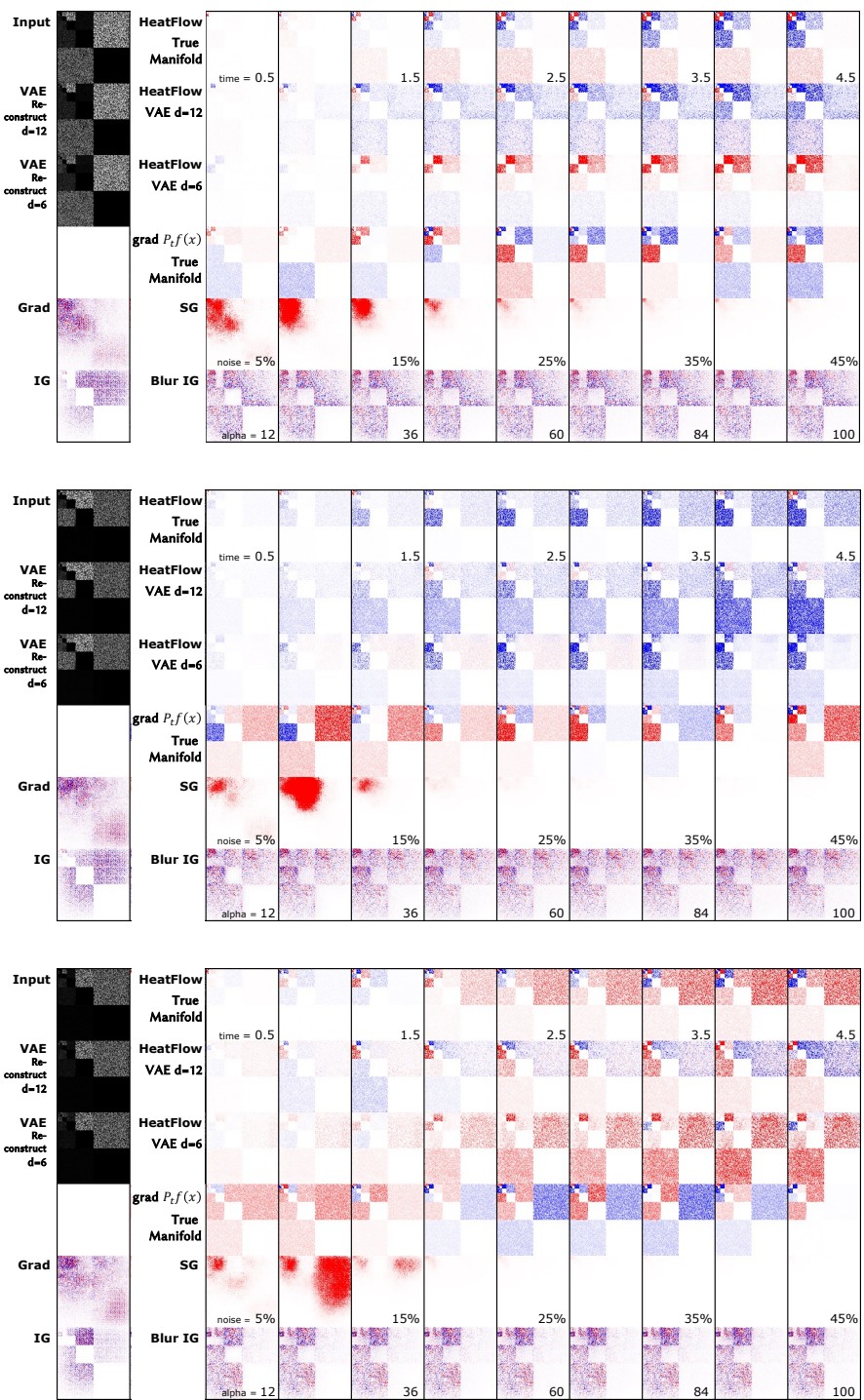

Figure 5: Three more examples for synthetic experiments. HeatFlow under true manifold, and learned manifold by VAE with latent dimension $d = 12$ and $d = 6$ are presented in the firt three rows, respectively. On the fourth row, gradients on the true manifold is collected at each time step. SG and Blur IG are presented on the last two rows.

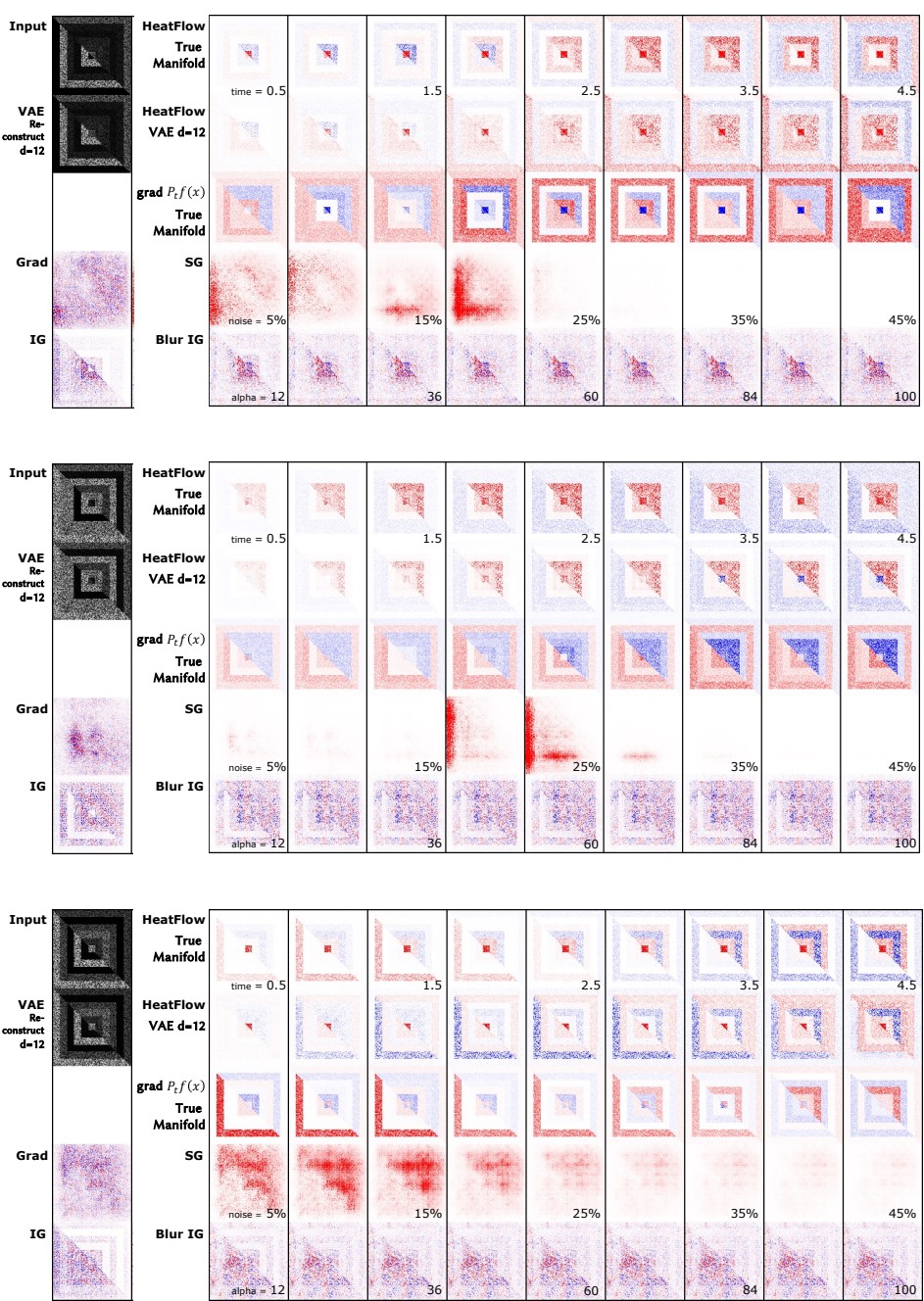

Figure 6: Three examples for another synthetic experiments. HeatFlow with true manifold, VAE with latent dimension $d = 12$ are presented in the first and second rows, respectively. On the thir row, gradients on the true manifold is collected at each time step. SG and Blur IG are presented on the last two rows.

## A.9 RESULTS FOR FACIAL AGE PREDICTION

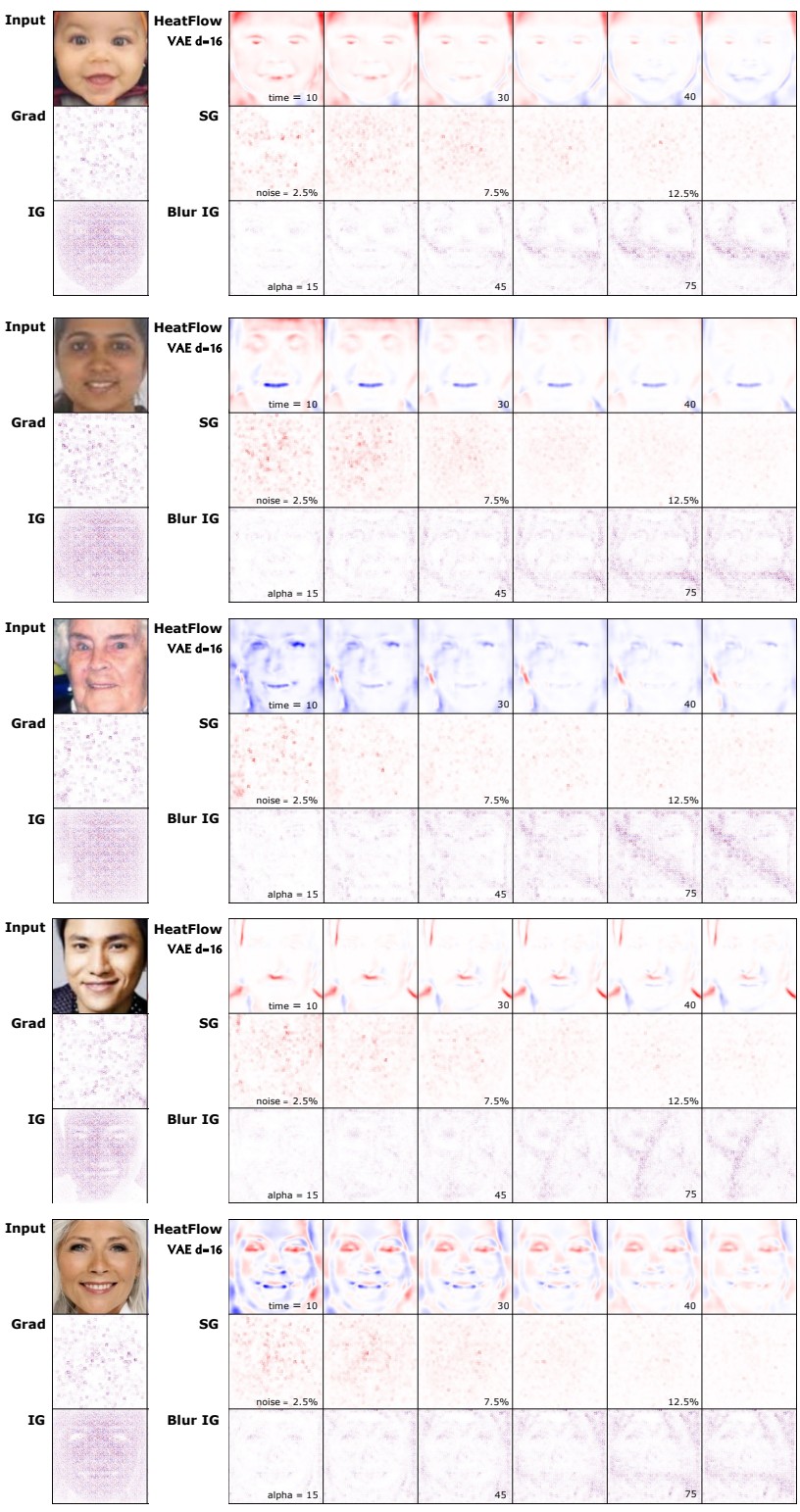

Figure 7: Five test examples for facial age prediction explanation. HeatFlow using VAE with latent dimension $d = 16$ are presented in the first row. Vanilla Grad and IG are shown below the original input. SG and Blur IG are presented on the second and thir rows.

A.10   ABLATION: COMPARING HEATFLOW ON MANIFOLD AND EUCLIDEAN SPACE

Figure 8: Four examples comparing HeatFlow on the learned manifold and Euclidean space $\mathbb{R}^{28\times28}$. Original input is shown on the most upper-left corner. HeatFlow run on the learned manifold and Euclidean space is shown on the first and second rows, respectively.

A.11   EXAMPLE OF ATTRIBUTION FOR ERRONEOUS PREDICTION

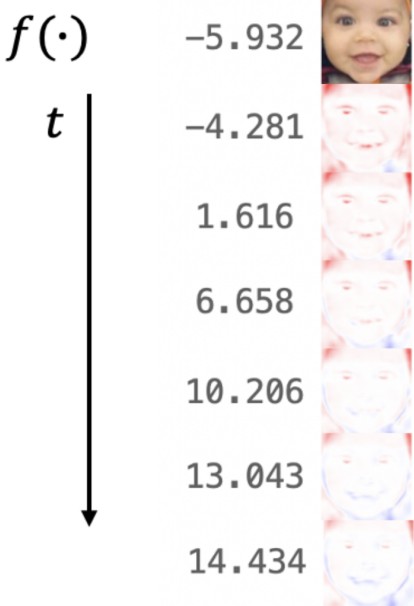

Figure 9: Example of attribution for an erroneous prediction from the facial age prediction task. The prediction of age of this example is $-5.9$, which is negative and wrong. As heat flows, indicated by a downward arrow, the prediction value of the smoothed model is presented beside each attribution result.

