# OpenReview forum: "Interpreting Neural Networks Through the Lens of Heat Flow"
_ICLR.cc/2023/Conference — Submitted to ICLR 2023_

### Official Review · Reviewer_Nuvr · 2022-10-23

**Confidence:** 4
**Correctness:** 2
**Technical Novelty And Significance:** 3
**Empirical Novelty And Significance:** 2
**Recommendation:** 3

**Clarity, Quality, Novelty And Reproducibility:**

Clarity: the paper is clearly written.

Quality: there are concerns about the evaluation strategy, as as things stand, the evaluation is limited and does not match up with state-of-the-art evaluation practices.

Originality: the idea builds upon the existing ideas of using heat equation, but it is applied in a different domain of interpretability of machine learning models, and therefore is novel. The efforts must be made to refer to existing machine learning approaches using heat equation (see Question 2).

Comments on quality:
1. Evaluation strategy can be improved in multiple ways. First, there is a doubt about the existing experiments. The first question would be : is MNIST a good dataset for evaluating interpretation? There is not much of a texture, and nearly any peak around the  contours of a symbol could be seen as a successful attribution. The results from the appendix, section A.9, make more sense to me. I think it would be great if the authors could comment on usability of MNIST dataset, maybe the reviewer misses something.  It is also important to show the legends for the heatmaps, see e.g. Figure 2 (which would clarify that say lower values are blue and the higher values are red). For Figure 4, it would be also important to include other datasets, including high-textured ones. Another question to answer is how it performs on the tasks related to the attribution, as it would help understand whether we can highlight specific features of the image (and also make it visually comparable to the other works). Simonyan et al, for example, use high-resolution dataset ILSVRC-2013. Xu et al (2020) use ImageNet and Diabetic Retinopathy data.  Also, do the predictions help identify causes of failures/what would the attribution look like for erroneous predictions?
2. Related literature: the paper discusses explanation methods, but it also needs to cover somehow the background of diffusion models and heat equations which has also recently been popular in the literature (see Kingma et al, 2021).
3. P5: increasing larger ts -> increasingly larger ts

**Strength And Weaknesses:**


Pros:
- It is an interesting and novel idea to use the diffusion equation for the attribution problem
- The paper is generally well written

Cons:
- Evaluation strategy and analysis (see comment 1 below)
- Related literature (see comment 2 below)


**Summary Of The Paper:**

The paper proposes a new neural network interpretation and attribution framework which looks at heat flow to achieve multi-scale interpretation of a scalar-valued function

**Summary Of The Review:**

The experimental evaluation needs to be more extensive, matching existing published models for interpretability and attribution (see Comment 1 above). Therefore, I do not recommend acceptance as things stand at this stage (but I welcome clarifications and improvements)

---

> ### Author Response · Authors · 2022-11-14
> **Reply**
>
> > is MNIST a good dataset for evaluating interpretation? There is not much of a texture, and nearly any peak around the contours of a symbol could be seen as a successful attribution. I think it would be great if the authors could comment on usability of MNIST dataset, maybe the reviewer misses something.
>
> MNIST is not uncommon in XAI literature:
>
>
> - Ancona, Marco, et al. "Towards better understanding of gradient-based attribution methods for Deep Neural Networks." International Conference on Learning Representations. 2018.
> - Smilkov, Daniel, et al. "Smoothgrad: removing noise by adding noise." arXiv preprint arXiv:1706.03825 (2017).
> - Sturmfels, Pascal, Scott Lundberg, and Su-In Lee. "Visualizing the impact of feature attribution baselines." Distill 5.1 (2020): e22.
> - Jethani, Neil, et al. "Have We Learned to Explain?: How Interpretability Methods Can Learn to Encode Predictions in their Interpretations." International Conference on Artificial Intelligence and Statistics. PMLR, 2021.
>
> MNIST is indeed an easy task for IG. As illustrated in Figure.3a, IG performs very well using an all-zero baseline input since background pixels always receive zero attributions. However, in regard of **multi-scaleness**, both smooth gradient and blur IG fail to produce meaningful multi-scale attributions especially when the scale of _bluring_ is large. But for HeatFlow, the attribution is still meaningful and endowed with the designed purpose because it always focuses on these digits and highlights different parts of the digits as the time goes.
>
>
> > It is also important to show the legends for the heatmaps, see e.g. Figure 2 (which would clarify that say lower values are blue and the higher values are red).
>
> Thanks for pointing it out. We had mentioned in the beginning of Section 5 in the original submission, that ``For all attribution maps of following figures, red and blue pixels denote
> positive and negative values, respectively.`` To make it clearer, we have also emphasized that ``deeper color denotes higher absolute value and stronger contribution`` in the revised submission.
>
>
>
> > For Figure 4, it would be also important to include other datasets, including high-textured ones. ... Simonyan et al, for example, use high-resolution dataset ILSVRC-2013. Xu et al (2020) use ImageNet and Diabetic Retinopathy data.
>
>
> UTKFace is already a relatively textured and high-resolution dataset, of size $200\times 200\times 3$, and showing detailed information about the age of human face, including wrinkles, smiles, eyebrows, etc. We recommend applying HeatFlow on the data manifold rather than Euclidean space to get more visually-appealing attributions (see Figure 8 in the Appendix), but to the best of our knowledge, currently there is no effective manifold learning techniques that work on datasets matching the scale and complexity of ImageNet.
>
>
> > Another question to answer is how it performs on the tasks related to the attribution, as it would help understand whether we can highlight specific features of the image (and also make it visually comparable to the other works).
>
> Comparison with other works on attribution task is exactly what we did for synthetic datasets, Mnist and UTKFace in Figure 3,5,6&7. For instance, on UTKFace the task is a regression one, where we predict age given a facial image. The attributions produced by our method clearly highlight specific features of the image that would bring up(down) the prediction of age if the test input to be predicted is below(above) the average age (~around 50 years old).
>
>
> > Also, do the predictions help identify causes of failures/what would the attribution look like for erroneous predictions?
>
> We have added an example of erroneous prediction from the facial age prediction task along with its multi-scale attribution results. Please refer to Appendix A.11.
>
>
> > Related literature: the paper discusses explanation methods, but it also needs to cover somehow the background of diffusion models and heat equations which has also recently been popular in the literature (see Kingma et al, 2021).
>
> Thank you for pointing this out. We have added the discussion of diffusion generative modeling in Section 4. As you have noticed, certain diffusion models (e.g., the variance-exploding SDE in Song, et al. with a linear schedule) are also based on heat equation. In our work, the heat diffusion is employed to smooth a given function $f$, rather than the implicit data distribution. Our work mainly relies on the Feynman-Kac represention of heat equation and was not motivated from the line of work on generative modeling (though we know the literature well).

---

> > ### Comment · Reviewer_Nuvr · 2022-11-15
> > **Still having some concerns**
> >
> > **"MNIST is indeed an easy task for IG. As illustrated in Figure.3a, IG performs very well using an all-zero baseline input since background pixels always receive zero attributions. However, in regard of multi-scaleness, both smooth gradient and blur IG fail to produce meaningful multi-scale attributions especially when the scale of bluring is large. But for HeatFlow, the attribution is still meaningful and endowed with the designed purpose because it always focuses on these digits and highlights different parts of the digits as the time goes."**
> >
> > That makes sense; however, it seems to me that there is a gap in the analysis the proposed method, which arises from that effectively we have only two evaluation tasks: the easy one which is MNIST, and the more complex one which is UTKFace. I wonder if similar evaluation is not feasible for ImageNet, maybe miniImageNet and CIFAR10/100 could help highlight the method's performance. Another possible way is to  use a larger-scale datasets with Euclidean space (perhaps in the appendix, if the limits of the paper do not allow it). It would be also important to highlight explicitly the following statement in the paper as it does not look like it can be directly understood from the text: **"to the best of our knowledge, currently there is no effective manifold learning techniques that work on datasets matching the scale and complexity of ImageNet."** It is not a problem per se that the method has limitations, and stating these limitations could improve the paper. Without the additional experimental results, both in terms of state-of-the-art attribution methods and diverse datasets, the paper may give incomplete empirical justification while being aimed at empirically-motivated task of interpretation of neural network predictions, and that is my biggest question.

---

> > > ### Author Response · Authors · 2022-11-16
> > > **Conclusions section has been revised**
> > >
> > > Thanks for your quick response.
> > >
> > > > I wonder if similar evaluation is not feasible for ImageNet, maybe miniImageNet and CIFAR10/100 could help highlight the method's performance.
> > >
> > > We have tried our method on datasets beyond MNIST for classification tasks but observed a particularly interesting phenomenon. When the inter-class similarity is low (e.g., airplanes vs cats in CIFAR10), even though it is possible to learn a VAE with acceptable reconstruction error, the VAE seems to fail to reveal a smooth underlying structure of the dataset. That is, we found that a random walk easily got stuck at some location. As a consequence, our method struggles at the PDE solving step. We conjecture that such dataset is more likely to be well modeled as a union of disconnected manifolds, rather than a single connected manifold.
> > >
> > > Hence, currently we recommand using the proposed method on data where the underlying manifold is a more connected one such as MNIST and UTKFace. It remains interesting to generalize HeatFlow to the union of disconnected manifolds, which may be a more reasonable assumption for classification datasets. To make the limitations of our work more apparent, we have summarized the discussion here in the Conclusion part in the revised submission and thank you for pointing them out.

---

### Official Review · Reviewer_s9nq · 2022-10-25

**Confidence:** 3
**Correctness:** 3
**Technical Novelty And Significance:** 3
**Empirical Novelty And Significance:** 2
**Recommendation:** 5

**Clarity, Quality, Novelty And Reproducibility:**

The idea is interesting to use the diffusion process in explainability. But the writing of this paper is hard to follow and the connection between the math symbol and explainability task should be added.

**Strength And Weaknesses:**

Pros:
The proposed FeatFlow is proved to be an additive model and satisfies the four axioms of the Shapley value.
This proof provides solid theory support that FeatFlow can provide fair feature attributions.

Cons:
This paper is a little hard to follow.
Although the authors define each symbol,
it's still hard to connect these abstract symbols to the explainability task.

For example, in section 2.1,
I think it can be better if more details about the Laplacian-Beltrami operator are given and provided some sample examples to show the relationship or motivation to apply this method on explainability.

A detailed pseudo-algorithm may be a better way to show the methodology clearly.

Question：
In Figure 4, it seems like IG gets the best result on the MNIST exclusion curve and it's better than the proposed HeatFlow.
However, in the UTKFace exclusion curve, it seems like IG has a much lower MAE than heat and SG.
Could you provide some insight into this phenomenon? I am curious about this.


**Summary Of The Paper:**

In this paper, the author proposes the HeatFlow.
It calculates the feature attribution using the model's outcome and local average values.
It provides multi-scale explainability results.

**Summary Of The Review:**

Above all, I think this paper provides an interesting framework to apply the PDE method to the explainability task.
HeatFlow can provide a fair attribution assignment by obeying four axioms in Shapley value.
However, the paper is hard to follow, and more details should be included to help the readers to understand.
I am willing to raise my score if the paper flow becomes easier to follow.

---

> ### Author Response · Authors · 2022-11-14
> **Reply**
>
> > This paper is a little hard to follow. Although the authors define each symbol, it's still hard to connect these abstract symbols to the explainability task. For example, in section 2.1, I think it can be better if more details about the Laplacian-Beltrami operator are given and provided some sample examples to show the relationship or motivation to apply this method on explainability
>
> Thank you very much for your suggestions. We have reorganized the flow of the paper, especially the Preliminaries part. The attribution problem is firstly introduced. After bringing in the motivation of defining a sequence of baselines which is the average model output evaluated on increasingly larger neighborhoods of the input to be explained, we then introduce the Lalplacian-Beltrami acting as a measure of deviation from local average, showing its natural connection to model explanation.
>
> > A detailed pseudo-algorithm may be a better way to show the methodology clearly.
>
> We have added a pseudo-algorithm showing a clear flow of the methodology. Please refer to end of Appendix A.5 in the revised submission.
>
>
> >  In Figure 4, it seems like IG gets the best result on the MNIST exclusion curve and it's better than the proposed HeatFlow. However, in the UTKFace exclusion curve, it seems like IG has a much lower MAE than heat and SG. Could you provide some insight into this phenomenon? I am curious about this.
>
> One inherent property of IG is that it will assign zero attribution to a feature if its value does not differ between the test input and the baseline. On MNIST, IG with an all-zero baseline always assigns zero contributions to these black background pixels by design. It means that IG can perfectly identifies these white pixels, as illustrated in Figure 3(a). On UTKFace, it seems that HeatFlow identifies key features much better than IG (see examples in Figure 7).

---

> > ### Comment · Reviewer_s9nq · 2022-11-26
> > **Further concern about the model-agnostic**
> >
> > Thank you for your hard work.
> >
> > Section 2.1 now clearly shows that FeatFlow can be applied as an importance assignment algorithm and provide the relationship to the current attribution methods.
> > The provided pseudo algorithm clearly shows that this algorithm contains two parts.
> > The first step is to train the VAE model to obtain the manifold, and then apply the HeatFlow algorithm to explain the importance of input features.
> >
> > However, I am confused about the relationship between the HeatFlow and the manifold training part.
> > Current attribution methods such as IG are usually model-agnostic algorithms, and they can be applied to explain models without specific requirements on the model architecture and training.
> > I am not sure whether the proposed HeatFlow can be treated as a model-agnostic algorithm.
> > If not, what's the reason such manifold training is required for this algorithm?

---

> > > ### Author Response · Authors · 2022-11-27
> > > **Model-agnostic**
> > >
> > > Thanks for your reply. Your previous suggestion helped improve the flow of our paper a lot.
> > >
> > > To first answer the question, the proposed HeatFlow is a model-agnostic algorithm as long as the explained model is a bounded continuous function, which is quite a general condition. This ensures the solution of the heat equation exists. We state a stronger condition (bounded and twice continuously differentiable) in the paper purely for ease of understanding. In comparison, IG relies on the [gradient theorem](https://en.wikipedia.org/wiki/Gradient_theorem), which requires that the function is a differentiable one. A continuous function is not necessarily differentiable, but a differentiable function is necessarily continuous. So our restriction on applicable models is even weaker in this sense. Intuitively, the convolution or expectation representation of the heat equation solution requires only zeroth-order oracle access to the model.
> > >
> > > As we have pointed out in the **manifold hypothesis** part (now moved to Section 2.1 in the revised version), the analysis can be drawn from the Euclidean space or restricted to the data manifold depending on the purpose of the model interpretation. If the goal is to understand model behavior in Euclidean space (which is implicitly assumed by many other explanation methods including IG and SG), no manifold learning step is required and one can directly start from the heat equation solving step by simulating Brownian motion in Euclidean space (i.e., adding Gaussian noise). Even when the data manifold is considered, the manifold learning step is required to be run only once and the learned manifold is reusable for any following model to be explained, as this step encodes information about data distribution regardless of models.
> > >
> > > Since usually a trained model is expected to behave consistently only on a low-dimensional submanifold, we find that constraining the analysis to the data manifold is key to the success of our method in getting more visually appealing attribution even when considering a large neighborhood. A comparison of choosing data manifold and Euclidean space on MNIST was illustrated in Appendix A.10. Another reason to suggest on-manifold HeatFlow is that, as shown in [1,2], off-manifold model evaluation should be avoided as much as possible for explanation methods because it may be manipulated arbitrarily.
> > >
> > >
> > > [1] Dylan Slack, et al. Fooling LIME and SHAP: Adversarial Attacks on Post hoc Explanation Methods. AIES 2020
> > >
> > > [2] Christopher J. Anders, et al. Fairwashing Explanations with Off-Manifold Detergent. ICML 2020

---

> ### Author Response · Authors · 2022-11-18
> **Reply**
>
> We are sorry to bother you. We hope our revision makes the idea easier to follow and our response resolves your main concerns. We would be grateful if you could consider updating the rating. Please let us know if you have any further questions.

---

### Official Review · Reviewer_uXwu · 2022-10-25

**Confidence:** 4
**Correctness:** 3
**Technical Novelty And Significance:** 1
**Empirical Novelty And Significance:** 1
**Recommendation:** 3

**Clarity, Quality, Novelty And Reproducibility:**

Clarity: Good. Quality: Fair; Novelty: Fair; It is a theoretical paper. I do not check the reproductivity. But the proof is right but the results are somehow standard in math community.

**Strength And Weaknesses:**

Strength: This paper is clearly written and well-organized.
Weakness: The paper introduces many math notations, while no enough emphasis on why heat flow can help us better understand the functions represented by neural networks for instance. There is no really crucial part which help us advance the understanding of DNN for instance. The results showed here are too toy models.

**Summary Of The Paper:**

This paper under review uses heat diffusion process to understand some multi-scale behavior of the learned model around a test point. Summary scales which characterizes the model on different scales can be drawn.

**Summary Of The Review:**

I would not suggest to accept this paper. It would be better developed before resubmit to a journal or a conference proceeding.

---

> ### Author Response · Authors · 2022-11-14
> **Reply**
>
> Thanks very much for your constructive feedback.
>
> >  The paper introduces many math notations, while no enough emphasis on why heat flow can help us better understand the functions represented by neural networks.
>
> To make the paper easier to follow, we have reorgnized the Preliminaries section and the model attribution problem and manifold hypothesis are firstly introduced. We have also added the intuitive connections between the Laplacian and the explanation problem.
>
> In addition, we have added a table in the Appendix A.7 to present the extrinsic definition of involved math symbols and illustrate them in the special case of Euclidean space for ease of understanding.
>
>
> >  There is no really crucial part which help us advance the understanding of DNN.
>
> We respectfully disagree with this assessment.
>
> - In XAI research and practice, attribution is widely recoganized as a meaningful task that helps identify the important features relied on by the model. Of course attribution methods cannot give us a complete understanding of how a complex model works in detail, but they are currently one of few active attempts to build reliable, human-friendly tools that help interpret general ML models in a mostly model-agnostic manner. The proposed method provides a theoretically sound approach to the attribution of differentiable models such as DNNs.
> - Compared with previously proposed attribution methods for differential models, our work suggests a more natural and robust target, $f - E_t f$, for attribution, shows how it is related to the Laplacian, presents how to compute the contribution of each feature to this difference on specified manifold, and proves the desired properties when applied in Euclidean space. We believe that our work is an interesting addition to the current family of ML model attribution techniques.
> - It is theoretically guaranteed that our work helps the understanding of DNNs (to some extent). For instance, as discussed in the paper, one consequence of Proposition 3 is that HeatFlow can help detect if a feature is contributing additively in a DNN. Combining Proposition 2 & 3, HeatFlow helps detect and recover the possibly additive structure presented in a DNN.
>
>
> >  The results showed here are too toy models.
>
> We argue that to be trustworthy, an explanation method should at least work on toy models. Unfortunately, most existing explanation methods have fundamental defects that prevent them from generating meaningful explanations even for toy models in some cases. For example, the gradient is always zero at critical points, which is generally not the case for HeatFlow (see Appendix A.3 where the attribution for the mean of Gaussian is analytically calculated). As another example, integrated gradients are highly sensitive to the choice of baseline, and a feature will always receive zero attribution if its value does not differ between the baseline and test point. Given these pitfalls, it is questionable whether the explanations for non-toy models output by these more scalable methods can be regarded as reliable.
>
> On the other hand, we acknowledge that the scalability of HeatFlow is inferior to widely used gradient-based explanation methods. We believe that our method will benefit from the methodological advancement of high-dimensional PDE solving and manifold learning, as well as the increasing availablity of computational resources.

---

> ### Author Response · Authors · 2022-11-18
> **Reply**
>
> Sorry for bothering you. We hope the revised paper is easier to follow and our response makes the contribution of this work clearer. Please let us know if you have any further concerns. We would be grateful if you could consider updating the rating.

---

### Official Review · Reviewer_VgMK · 2022-10-25

**Confidence:** 1
**Correctness:** 3
**Technical Novelty And Significance:** 3
**Empirical Novelty And Significance:** 3
**Recommendation:** 6

**Clarity, Quality, Novelty And Reproducibility:**

Clarity:

The paper is a little dense and hard to read. It might be because I am not familar with the math used in this paper.

Quality:

The mathematical background used in this paper seems to be solid.

Novelty:

To me, the use of the heat equation on a Riemannian manifold to interpreting deep learning models is novel. However, I do not know modern literatures on this field well-enough to evaluate correclty.

Reproduciblity:

The paper contains an incomplete set of used model architectures and hyper-parameters. Experimental code is not made publicly available (although it might be not a ciritcal issue considering this paper is theoretical one).


**Strength And Weaknesses:**

Disclaimer:

 I am not an expert of the model interpretaion and explainable AI fields. My evaulation on this paper is not exhaustive and might be incorrect.

Strength:

1. The paper is based on solid theoretical properties from differential geometry and PDE. It seems that the proposed method is technically sound.

2. Experimental results show that the proposed HeatFlow is likely to interprete the model more convincely compared to other baselines.

Weakness:

1. The proposed method can only address a scalar case, i.e., 1D regression problem or binary classification. However, it can be by-passed by investigating each output of multi-class/multi-dimensional models separately.

2. The paper is a little dense and hard to read. It might be because I am not familar with the math used in this paper.



**Summary Of The Paper:**

The authors propose HeatFlow, a framework based on the heat equation on a Riemannian manifold, to interpreting a representation of givn deep learning model. Especially, they focus to analyze the multi-sale behvariors of the model, which cannot be revealed by using a naive gradient-based model interpretation approach. The authors show that the proposed HeatFlow framework obeys some nice properties that frequently used in the interpretable AI field, e.g., attribution axioms. The authors compare the proposed method with four baselines including Grad, IG, SG and BlurIG

**Summary Of The Review:**

While it is an educational guess, I think this paper contributes both a solid theoretical background and practial method for the relevant field. I would like to vote to accept this paper. However, since I am not an expert in this field, I cannot evaluate this paper exhaustively; thus, my rating might be updated after discussing with the authors and other reviewers.

---

> ### Author Response · Authors · 2022-11-14
> **Reply**
>
> Thanks very much for your supportive review.
>
> > The paper is a little dense and hard to read. It might be because I am not familar with the math used in this paper.
>
> To make the paper easier to follow, we have reorgnized the Preliminaries section and the model attribution problem and manifold hypothesis are firstly introduced. We have also added the intuitive connections between the Laplacian and the explanation problem.
>
> In addition, we have added a table in the Appendix A.7 to present the extrinsic definition of involved math symbols and illustrate them in the special case of Euclidean space for ease of understanding.
>
> In a nutshell, HeatFlow in its simplest form (i.e., in Euclidean space) attributes $HeatFlow_i(\mathbf{x}, T) = \int_{0}^T \partial^2 \mathbb{E}[f(X_t) | X_0 = \mathbf{x}] / \partial x_i^2 dt$ to the feature $i$ as its contribution to the deviation $f(\mathbf{x}) - \mathbb{E}[f(X_T) | X_0 = \mathbf{x}]$. This attribution has several nice properties as proved in Section 3.3, in which the most important one is $f(\mathbf{x}) - \mathbb{E}[f(X_T) | X_0 = \mathbf{x}] = \sum_{i=1}^D \text{HeatFlow}_i (\mathbf{x}, T)$. As presented in the paper, it is possible to preserve this property on a general submanifold through disaggregating the more general Laplace-Beltrami operator.
>
> > Experimental code is not made publicly available.
>
> The link to anonymized source code was provided in the footer of page 7: https://anonymous.4open.science/r/heat-explainer-FFD0

---

### Decision · Program_Chairs · 2023-01-20

**Decision:**

Reject

**Justification For Why Not Higher Score:**

The reviewers are not convinced by the technical and empirical results in this paper. The writing also lacks clarify, and the experimental results are not comprehensive. Thus a reject is recommended.

**Justification For Why Not Lower Score:**

NA

**Metareview: Summary, Strengths And Weaknesses:**

This paper interprets neural nets by using heat diffusion process. The reviewers are not convinced by the technical and empirical results in this paper. The writing also lacks clarify, and the experimental results are not comprehensive. Thus a reject is recommended.